# StateSpaceDiffuser: Bringing Long Context to Diffusion World Models

**Nedko Savov**[1,†]     **Naser Kazemi**[1]     **Deheng Zhang**[1]     **Danda Pani Paudel**[1]
**Xi Wang**[1,2,3]     **Luc Van Gool**[1]

[1] INSAIT, Sofia University "St. Kliment Ohridski"     [2] ETH Zurich     [3] TU Munich

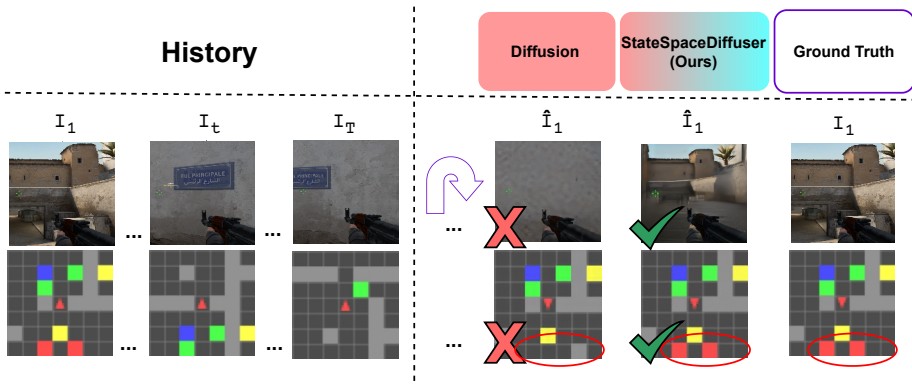

Figure 1: **Recalling content long in the past.** Given a history of images $I_1, ..., I_T$ and accompanying actions, we navigate all the way back to the beginning - $I_1$. The task is to generate frames along the way, consistent to what is seen in the history, given the actions. As an example, we show the predictions of the first frame - $\hat{I}_1$. Can a generative model recall the content of $I_1$ long back in the sequence? Diffusion models fall short (✗), our model correctly recalls the content of $I_1$ (✓).

## Abstract

World models have recently gained prominence for action-conditioned visual prediction in complex environments. However, relying on only a few recent observations causes them to lose long-term context. Consequently, within a few steps, the generated scenes drift from what was previously observed, undermining temporal coherence. This limitation, common in state-of-the-art world models, which are diffusion-based, stems from the lack of a lasting environment state.

To address this problem, we introduce StateSpaceDiffuser, where a diffusion model is enabled to perform long-context tasks by integrating features from a state-space model, representing the entire interaction history. This design restores long-term memory while preserving the high-fidelity synthesis of diffusion models.

To rigorously measure temporal consistency, we develop an evaluation protocol that probes a model's ability to reinstantiate seen content in extended rollouts. Comprehensive experiments show that StateSpaceDiffuser significantly outperforms a strong diffusion-only baseline, maintaining a coherent visual context for an order of magnitude more steps. It delivers consistent views in both a 2D maze navigation and a complex 3D environment. These results establish that bringing state-space representations into diffusion models is highly effective in demonstrating both visual details and long-term memory. Project page: https://insait-institute.github.io/StateSpaceDiffuser/.

---

[†]Corresponding author: nedko.savov@insait.ai

39th Conference on Neural Information Processing Systems (NeurIPS 2025).

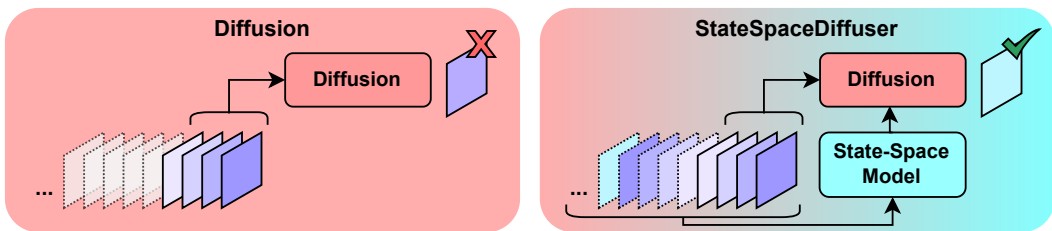

Figure 2: **Our Approach.** While diffusion models are limited to a short sequence input, our approach enables long-context processing for diffusion models with a state-space representation.

# 1 Introduction

World models have gained popularity for the production of visual consequences of given past observations and actions. These models can learn to generate environment observations entirely by training on many interactions with the environment. Simply by observation, they are capable of handling complex environments, such as car driving [27, 29, 41, 66], 3D virtual environments [77, 1, 19], platformer games [8, 68], ego-centric action videos [82], or navigation [2, 84]. They enable interactivity without the burden of hand-coding complex environments, but also offer feature representations for robotics and reinforcement learning agents for planning.

For long interaction with world models, it is essential that the generated video remains consistent with previously observed or generated content. Revisited areas should preserve their appearance, and objects observed again should keep their properties. However, as shown in Fig. 1, current high-fidelity world models, which are mostly based on diffusion, cannot preserve context outside of a short time window, most often directly limited by their input window size [82, 1, 19, 8]. This leads to an increasing drift in content over time, where earlier information is forgotten or overwritten. The inability to retain persistent memory of the environment poses a major challenge, especially for real-world applications such as agent planning and virtual interaction, where coherent, temporally consistent environments are essential. Therefore, in this work our task is to stay consistent with a long history of past inputs, even if generating a single future frame. This is in contrast to long generation which focuses on producing extended realistic sequences, even without prior context.

To improve content consistency in diffusion-based world models, we make use of a persistent long-context representation. Specifically, we leverage features from a discrete state-space model (Mamba), which has been shown to be very effective at capturing long-term context in prior work [20, 31]. We summarize our system in Fig. 2. Although these models were previously applied to language and visually simple environments [12], our goal is to preserve the long-term context in modern diffusion-based world models, targeting environments with higher visual complexity such as CSGO [60]. In contrast, other state-based models, such as those using LSTMs or GRUs [38, 39, 40], have limited generative capacity and are mainly used for agent planning. Our approach combines the strong generative power of diffusion models with the long-term context tracking of state-space representations.

Importantly, the state-space model (SSM) is computationally efficient, which allows it to process arbitrarily long sequences. This is achieved by maintaining a compact state that is updated at every sequence step. During training, SSMs have linear complexity in sequence length [31], further improved by parallelization. Unlike them, CNNs have a fixed receptive depth, and transformers characterize with a heavy quadratic complexity. At inference, in a streaming fashion, the SSM can be executed with constant per-step latency and constant memory footprint, whereas transformers and CNNs, at best, still grow linearly per step. As we show, in test time, our proposed model scales far beyond its trained horizon, while the SSM contributes less than 2% of the total inference compute of the full model.

Our proposed model, StateSpaceDiffuser, summarized in Fig. 2, consists of a state-space model that operates over the long sequence, and a diffusion model - conditioned on both a short window of observations and state-space model features. The latter enables the diffusion branch to generate the content of the next frame conditioned on a long context rather than the last few frames.

To evaluate the consistency of the generated long context of StateSpaceDiffuser, we design and develop an evaluation framework that involves navigating environments to then return back to the

initial position. We evaluate on two environments. (1) A simpler 2D maze environment (MiniGrid), in which we establish the presence or absence of memory ability by remembering the maze layout given partial observations. And (2) a 3D first-person shooter game (CSGO), which serves to show the performance of our method on a visually challenging interactive environment with many factors at play. Our quantitative and qualitative evaluation results show that StateSpaceDiffuser produces content significantly more consistent with a long history than a diffusion-only method. Evaluation in the maze environment yields 51.9% PSNR improvement over the baseline on average (56.3% improvement on the most memory challenging cases). A user study confirms that our method produces images closer to previously observed content in the CSGO dataset compared to baselines. More details are shown in Sec. 5.

Our contributions are as follows.

- We propose StateSpaceDiffuser, which integrates a state-space model with a diffusion model for visual world modeling. It is capable of generating consistent content in long-horizon generation, with almost no extra computational cost.

- We develop an evaluation protocol to test the content preservation abilities of a world model and perform extensive evaluations of world models on long-horizon generation tasks.

- Our evaluation shows a significant quantitative improvement and a strong user preference over the baseline in the case of long contexts. Furthermore, our studies attribute the improvements to our model design and confirm generalization to longer contexts.

## 2 Related Work

### 2.1 World Models

**Generative environment models.** Initially developed as imagination-based models for training model-based reinforcement learning (MBRL) agents [13, 38, 40, 70], world models have evolved into powerful generative systems that condition on actions to produce future frames [11, 41, 57, 83]. Early work by [36] demonstrates that training a recurrent latent dynamics on VAE image representations can enable agents to plan in imaginative rollouts. Extensions such as SimPle [47] and Dreamer [37] refine this approach by improving reconstruction quality and stability, culminating in DreamerV2 and DreamerV3 [39, 40] - systems that achieve human-level performance on Atari and demonstrate the ability to generalize across diverse domains. More recent efforts, such as IRIS [58], TWM [64], STORM [86], and DayDreamer [79], employ Transformer-based hybrid backbones and focus on sample efficiency, long-horizon coherence, or robotic control. However, many of these methods rely on discrete latent tokens and relatively short contexts, which limits visual fidelity in complex scene motion or when extended rollouts are required.

World models are also central to realistic video generation conditioned on actions. Genie [8] leverages a video tokenizer and a Latent Action Model for dynamic next-frame generation, whereas GAIA-1 [46], GAIA-2 [66] tackle autonomous driving by autoregressively predicting image tokens from multi-modal inputs. Recent works highlight broader applicability and complex generative capabilities. DINO-WM [87] uses pretrained visual features for zero-shot planning, GameFactory [84] adapts game environment actions to realistic environments, while allow video generation control by periodical text instructions. Both illustrate how world models can transcend traditional RL frameworks and support open-ended content creation.

**Diffusion-based approaches.** Parallel to these developments, diffusion models [73, 44, 75] have emerged as a powerful class of generative methods for high-fidelity image and video synthesis. They have been applied to text-to-video [71], space-time video generation [3], and broad world simulation tasks [6]. Within MBRL, DIAMOND [1] uses a diffusion model to generate high-quality frames for Atari, making for playable environments and enhancing agent performance. Methods like Pandora [80] and LCT [35] generate video based on periodic text instructions. Nonetheless, current diffusion-based world models, typically transformer-based, condition on only a short window of past frames to handle the quadratic computational complexity, making long-horizon dependencies difficult to maintain. This makes it challenging to maintain long-horizon dependencies.

## 2.2 Sequence Modeling

**RNNs and Transformers.** Sequential modeling has historically been dominated by recurrent neural networks (RNN) such as LSTM and GRU [45, 15, 17], which process input tokens step by step and are able to handle moderate contexts. However, RNNs often struggle with extremely long sequences due to vanishing gradients and limited memory capacity [59]. Transformers [78] addressed these issues by employing self-attention, making them effective in capturing long-range dependencies. Beyond world modeling, Transformers have become the backbone for a broad range of tasks, including language modeling [21, 7, 63] and computer vision [22, 43, 9, 28], due to their ability to handle global context. Various Transformer variants have attempted to reduce the quadratic cost of self-attention for long sequences [50, 16, 4, 85, 14]. Vision-specific models like Swin [56] or MViT [24] adopt hierarchical or local attention, yet scaling them to long video horizons remains computationally prohibitive.

Previously, DFoT [10] addressed the ability for long future prediction. However, the long-context consistency problem has only been recently addressed by a few concurrent works. [51, 74, 81] improve context abilities by proposing strategies to sample a number of historical observations to use as conditioning. Instead, our approach involves summarizing information from the entire history automatically through state-space models.

**State-Space Models (SSMs).** As an alternative, SSMs [5, 53, 62, 76, 61] can process sequences in linear time by learning continuous dynamics in a latent state. Representative structured state-space models include S4 [33, 34] and H3 [18] that generalizes the recurrence in Linear Attention [49]. S4, S5 [72], and S6 [52] leverage carefully designed operators (e.g., HiPPO matrices [32]) to efficiently capture long-range dependencies. Mamba [31] introduces selective gating to improve expressiveness without sacrificing linear scalability. S4WM [20] has shown that applying SSMs as world models shows promise for maintaining coherence over hundreds of imagined steps while preserving computational tractability.

**Hybrid Architectures.** As Transformers excel at local interaction with low computational cost and SSMs can capture long-horizon dependencies efficiently, hybrid designs have been proposed for vision tasks. MambaVision [42] incorporates state-space models into a transformer, and Dimba [26], DiS [25] - into a diffusion network backbone, for computationally cheaper image discriminative and generative tasks, operating on image patches.[54] modify the softmax in attention to emulate a forget gate and improve transformer context abilities. MambaVLT [55] and Samba [69] exploit state-space models for better object tracking with long-range consistency.

# 3   Data

We design an experimental protocol to evaluate long-term content consistency in diffusion world models, comprising of three experimental setups with a rising level of complexity, based on a controlled maze environment (MiniGrid) and a complex 3D first-person environment (CSGO).

We create a dataset based on the partially observed MiniGrid maze environment [12]. In this setup, each maze consists of a grid where each cell can be a wall, an empty space, or a colored marker. Markers act like empty spaces, but are visually distinct. An agent navigates the maze, but at each time step, it only sees a 7×7 window centered around itself rather than the full 85×85 maze (see Fig. 6 (b) for an example). We use a modified version of MiniGrid with randomly generated mazes, allowing us to adjust the size, wall complexity, and number of color-coded markers. In each episode, the agent is tasked with visiting a sequence of 40 random markers via the shortest path. Once halfway through the episode, the agent stops following the path and retraces its steps back to the starting point. Each episode is 100 steps long (50 forward, 50 backwards). We evaluate on different context lengths by selecting subsequences around the long sequence center. Notably, the second half of each sequence depends heavily on the model's ability to recall earlier frames, making it ideal for testing long-context reasoning.

We also design a simplified dataset called MiniGrid Simple, consisting of just 34 samples without walls and a single marker placed behind the starting position. The agent moves three steps forward and three steps back, returning to its initial position. Since the context window of our baseline is just 4 steps, this setup provides a minimal but effective test of long-term recall. We use this to compare the performance of our baseline and state-space-enhanced models in reconstructing the marker color.

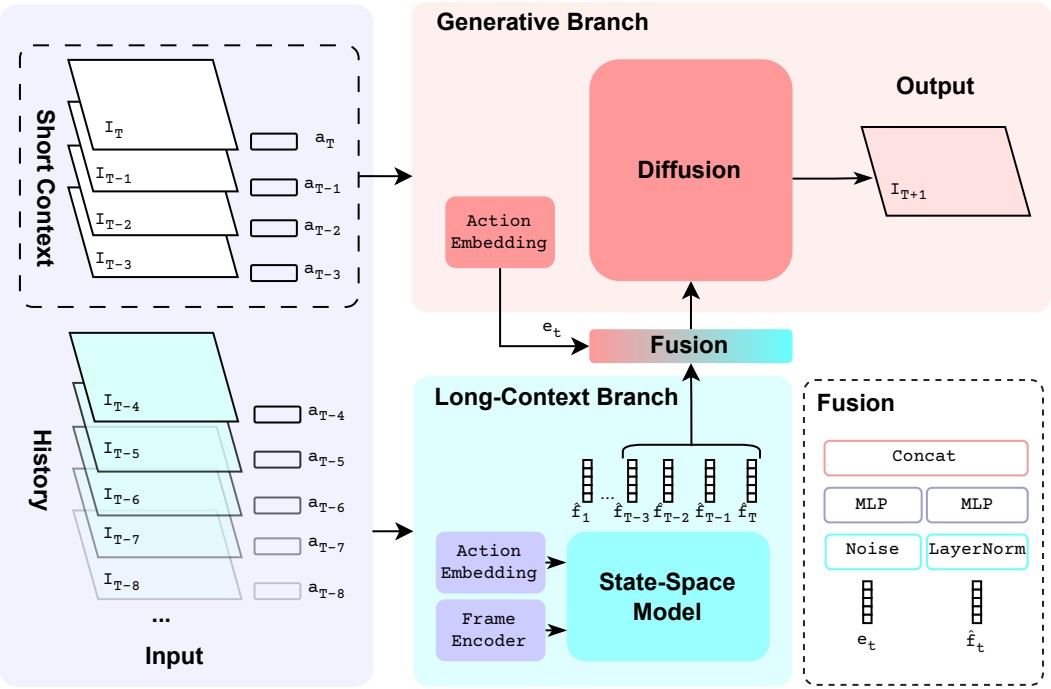

Figure 3: **Architecture of our StateSpaceDiffuser model.** It consists of: a state-space model for processing long context information; a diffusion model generating high-fidelity context-aware next observation, conditioned on state-space features.

To evaluate our method in a more visually complex setting, we use CSGO [60], a dataset of human gameplay in a 3D first-person shooter. It includes 51 action types, such as 23 rotational commands, 4 movement directions, jumping, and various special actions (e.g., firing, changing weapons). To adapt the dataset for long-context testing, we create mirrored sequences: for each original sequence, we append its reversed version, ensuring that actions are also reversed. We use a corresponding one-hot encoding (e.g. turning left becomes turning right), or create a new one if a correspondence does not exist (e.g. jump, shoot). This setup forces the model to rely on information from earlier in the sequence when generating later frames.

## 4 Methodology

Given a sequence of environment interactions $a_1, a_2, ..., a_{T-1}$, the resulting observations $I_1, I_2, ..., I_T$ (with initial frame $I_1$) and the current action $a_T$, the objective of a world model $\mathcal{F}(\cdot)$ is to produce the next image $I_{T+1} = \mathcal{F}([I_1, ..., I_T], [a_1, ..., a_T])$. Recently, the best-performing generative architectures for modeling $\mathcal{F}(\cdot)$ are diffusion models based on transformers or UNet. In training, transformers are computationally intensive - $O(T^2)$. CNNs have fixed receptive fields and are ill-fitted to long-term dependencies. Therefore, these models take a short history window of observations: $I_{T+1} = \mathcal{F}([I_{T-K+1}, ..., I_T], [a_{T-K+1}, ..., a_T])$. (e.g. $K = 4$[1, 77], $K = 16$[8]). With a long and growing sequence, the short history causes the loss of long-term temporal coherence. Instead, we propose to efficiently process the long sequence ($O(T)$) with a model designed for this purpose - a state-space model. Such models maintain and update a state with each sequence step, and the state serves as a summary of the sequence so far. Extracting long-context features in this way and integrating them into the diffusion pipeline yields the proposed model - StateSpaceDiffuser.

### 4.1 StateSpaceDiffuser Architecture

Our architecture is shown in Fig. 3. It is conceptually divided into two branches: Long-Context Branch and Generative Branch. The Long-Context Branch preserves information over long sequences, and the Generative Branch uses this context to render high-quality images.

**Long-Context Branch**    In contrast to transformer and CNN architectures, state-space models (SSMs) are designed specifically to efficiently process long sequences. Although SSMs are generally designed for continuous input signals, discrete SSMs maintain an internal state representation $h$ that is updated with each time step $t$ in an input sequence of one-dimensional feature vectors $f_1, ..., f_T$. , through the parameter matrices $A$, $B$ and $C$, which are learned in training time:

$$h_t = Ah_{t-1} + Bf_t, \quad m_t = Ch_t$$

We denote a state-space model with $m_1, ..., m_T = \mathcal{M}(f_1, .., f_T)$, with $m_t$ denoting the model's output. To bring it into the world model setting, we define $f_t$ to be a compact feature representation of $I_t$ and train a model that predicts future observations: $\hat{f}_2, ..., \hat{f_{T+1}} = \mathcal{M}([f_1, a_1], .., [f_T, a_T])$. It is common in existing work to apply SSMs at the patch or image token level as common in previous work [42, 26]. Instead, we avoid conflating spatial and temporal dependencies by temporally processing full frames. Each frame is encoded into a single compact feature $f_t$ used as a single step of our sequence. The encoding is obtained by the continuous Cosmos tokenizer [23] with scale 16 (CI16). The resulting patch tokens (dimension 16) are flattened to form the single feature vector $f_t$ per image. Alongside it, we incorporate a discrete action $a_t$, which indexes a learnable embedding of dimension 16. We concatenate $f_t$ and $a_t$ to create the input at each step. We adopt the Mamba architecture due to its dynamic selection mechanism and efficient parallelism, which led to superior performance compared to other SSM variants.

One key benefit of state-space models is their computational efficiency. As only a single state is maintained in inference, memory remains constant regardless of context length. As the same update is applied linearly on a sequence, computational complexity remains $O(T)$. When presented with a growing sequence, Mamba only updates the state from the previous step, making for a constant per-step latency. In contrast, CNNs and transformers do not maintain a state and have to reprocess the growing sequence at every step. In training, Mamba further parallelizes the sequence processing.

Provided a long input sequence of high-resolution images and corresponding actions, the model predicts the Cosmos features corresponding to the next observation with an MSE loss. In those features, we expect relevant to the time-step information, recalled from the sequence. While context cues are to be preserved, the state is low-dimensional, and this model is not generative. Therefore, the generative branch, designed for the generation of high-quality images, is intended to render the final output.

**Generative Branch.**    To generate high-quality images in complex environments, we employ a diffusion model. Our choice is the DIAMOND world model [1], a UNet-based EDM diffusion model [48] designed for visual prediction in sequential environments. Therefore, our Generative Branch conditions on only four low-resolution frames and their corresponding actions, represented as 512-dimensional action embeddings. Despite this minimal context, it can produce high-quality predictions with just three denoising iterations per output frame. The architecture consists of two diffusion models: a primary model that predicts the next observation at a low resolution, and a secondary upsampler that refines these predictions to a resolution of $280 \times 150$. As the model predicts one frame at a time, generating longer sequences is achieved through a sliding-window strategy, and each newly generated frame is appended to the input history for the next prediction. In isolation, this strategy causes a short-context limitation to this model.

**Fusion of features.**    To address the context limitation of the Generative Branch, we build a fusion module to integrate state-space features into it, in order to provide long-context information. To that end, we process the entire sequence with the Long-Context branch and obtain the last 4 output features $\hat{f}_t$. We fuse those features with the corresponding action embeddings from the Generative Branch. These features are first normalized and then passed through a two-layer MLP with SiLU activation, where the input size matches the feature dimensions. Similarly, the action embeddings, perturbed with noise, are processed by an MLP with the same architecture. To form the final conditioning vector, we concatenate the outputs of the two MLPs. Empirically, we discovered that processing the memory and action conditions independently before concatenation yielded better performance than fusing them earlier in the pipeline.

### 4.2 Training Protocol

At each step, a batch consists of sequences of actions, reversed actions and observations. Training is performed in two stages. Firstly, the Long-Context Branch is trained on long context - length 50 or 16. The produced features decode to images with artifacts, but with important context cues. Then, freezing the Long-Context Branch, we train the Generative Branch, conditioned on the compressed long-context features, with a sequence size of 4. This branch produces the final high-quality images with the correct context. Training details are given in App. A.

We found that this two-stage training is crucial for stability. Direct end-to-end training is unstable, as diffusion gives noisy gradients to the SSM, and the SSM gives constantly changing features to diffusion. In turn, diffusion learns to ignore the SSM features. Therefore, stable features of a pretrained SSM worked best in this architecture. Moreover, the training separation enabled to swap out in test time the Long-Context Branch with another independently trained model, without having to further fine-tune the heavier Generative Branch.

## 5 Experiments

### 5.1 Experimental Setup

**Baselines.** We establish two baselines. The first is a pure diffusion model without state-space features: the DIAMOND model. Our second baseline is the State-Space World Model. It is the Long-Context branch of StateSpaceDiffuser and its training is equivalent to the first stage of training, as described in Sect. 4.2. At inference time, the predicted feature $\hat{f}_t$ is decoded into an image $I_t$ using the decoder from the Cosmos tokenizer. In App. B.2 we present comparisons of sequence models to solidify our choice of Mamba as our backbone.

This model enables us to assess the memory capacity of state-space models (SSMs) in sequential visual prediction. Although its outputs tend to be blurry and contain artifacts in complex scenes, due to the absence of a variational component and limited generative expressiveness compared to modern diffusion models, the SSM exhibits a strong ability to model long sequences and retain information from earlier in the trajectory. The strengths and shortcomings observed in this baseline directly inform and motivate the design of StateSpaceDiffuser.

**Testing Protocol.** Our evaluation protocol matches our mirrored action setup - we take $n$ actions and $n$ reverse actions, and expect to generate the same observations for the second half of the sequence as seen in the first. On MiniGrid, we have a fixed sizeable visual difference per step, while for CSGO continuous motion often results in small per-step changes. Therefore, in MiniGrid, we generate one frame in the future at a time, while in CSGO we sequentially generate the whole second half of the sequence. On MiniGrid we evaluate with PSNR and SSIM on varying future horizons - the further in the sequence, the longer the memory required. In CSGO we perform a user study, more aligned to the visual complexity of the environment. We motivate this difference with the known mismatch between perceived quality and fidelity metrics in continuous video [65, 67, 30] (App. D.3). Although the baseline performs well when context is not essential, our protocol exposes its inability to model long-term context, resulting in degraded quality in this scenario.

### 5.2 Results and Analysis

**Simple MiniGrid Evaluation.** In this experiment, we test the recall ability of the baseline, the State-Space World Model and StateSpaceDiffuser, on a simple toy setup, as described in Sect. 3. We train and test on the same set of 34 samples. The goal is to recall a color at the final frame from the first frame in the sequence with a length of 7 frames. Two random samples (colors) from the results are shown in Fig. 4, with the corresponding model predictions. With input size 4, the baseline processes the sequence in a sliding window fashion and, as within the 3 steps the color information is lost, it cannot reconstruct the correct color. Despite the small training set size, the baseline fails because of a lack of long-context abilities. In contrast, our State-Space World Model, based on a computationally efficient state-space model, is able to predict the correct color. Finally, it is demonstrated that our StateSpaceDiffuser is also able to recall the correct content by effectively combining both paradigms. Notably, our methods perform equivalently on a context length of 50 frames - when predicting the 51st, StateSpaceDiffuser recalls the color from 50 steps ago.

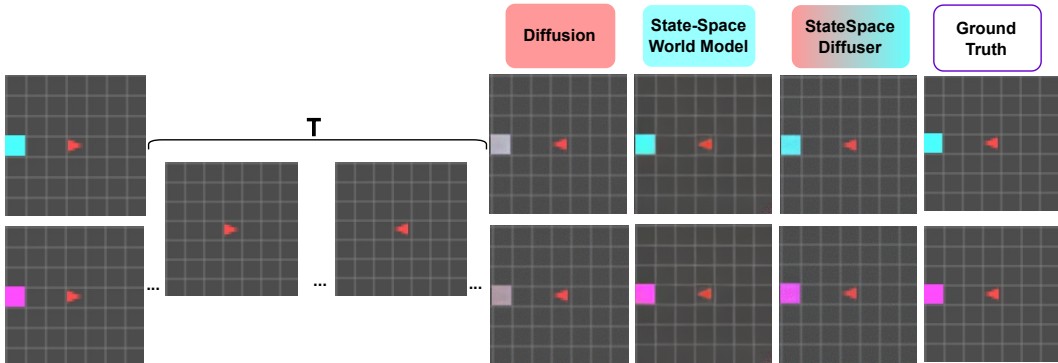

Figure 4: **Long-Context Simple Demonstration.** An agent starts moving to the right covering the color for the next T/2 frames, then the agent moves back the same amounts of steps. Diffusion baseline fails to recover the color, StateSpaceDiffuser and State-Space World Model successfully recover it through the state-space representation. Both T=7 and T=50 generates such results.

| Model | Avg. PSNR↑ | Fin. PSNR↑ | SSIM↑ |
|---|---|---|---|
| *Context Length 16* | | | |
| DIAMOND | 27.13 | 25.44 | 0.95 |
| State-Space World Model | 33.40 | 33.17 | 0.96 |
| StateSpaceDiffuser (Ours, w/o state) | 23.68 | 20.95 | 0.92 |
| **StateSpaceDiffuser (Ours)** | **41.01** | **40.55** | **0.98** |
| *Context Length 50* | | | |
| DIAMOND | 26.13 | 25.15 | 0.95 |
| State-Space World Model | 32.64 | 32.44 | 0.96 |
| **StateSpaceDiffuser (Ours)** | **39.68** | **39.32** | **0.98** |

Table 1: **MiniGrid Quantitative Evaluation of Long-Context Awareness.** Our StateSpaceDiffuser outperforms the baselines.

Figure 5: **CSGO User study results.**

**Forward-Backward Evaluation on MiniGrid.** We compare the long context abilities of our diffusion (DIAMOND) and state-space (State-Space World Model) baselines in our MiniGrid test set. We evaluate our models trained on context length 50 on context lengths 16 and 50 (demonstrating generalizability). We follow the protocol outlined in Sect. 5.1. To evaluate, we compute the Peak Signal-to-Noise Ratio (PSNR) for each predicted frame in the reverse trajectory, reporting both the mean score and the PSNR at the final time step, which requires the longest-term memory. As shown in the Tab. 1, our model significantly outperforms both baselines, particularly at the end of the sequence, where successful recall of the first frame is critical. This highlights the model's ability to retain and reinstantiate long-term visual context. In App. B.4, B.5, we show the stability and robustness of these results, in App. B.1 - performance gain analysis over computational cost. Compared to the State-Space World Model, our method achieves higher fidelity output, benefiting from the superior generative capacity of the Generative Branch (examples - in App. C.1, C.2). Fig. 6 (b) presents example rollouts generated by our model and the diffusion-only baseline. In MiniGrid, predictions are made one step at a time using the ground truth sequence. As a result, most content is carried over from the previous frame, with only the newly revealed area requiring inference. Our method excels at filling in these newly revealed regions, even when the relevant context originates far back in the sequence. In contrast, the diffusion baseline struggles to recover such long-range dependencies.

**Recall Across a Context Length.** In this experiment we study the accuracy of our models over the varying context length of the forward-backward evaluation on MiniGrid. When predicting future observations, the last frame's content depends on the first frame's content, and the further back we go in the sequence the smaller the context length required for a good reconstruction. This is a direct consequence of the mirror style of the observations in our setup. In Fig. 7 we show the PSNR at each predicted time step. The first few predicted frames are easily predicted by all models as the solution falls within the short input window. However, performance for the diffusion baseline quickly falls as no form of information is preserved from the long context, while a state-space model is able to harvest this information. Our StateSpaceDiffuser model gets the best of both worlds - long-context awareness and high-fidelity predicted images, and performs the best.

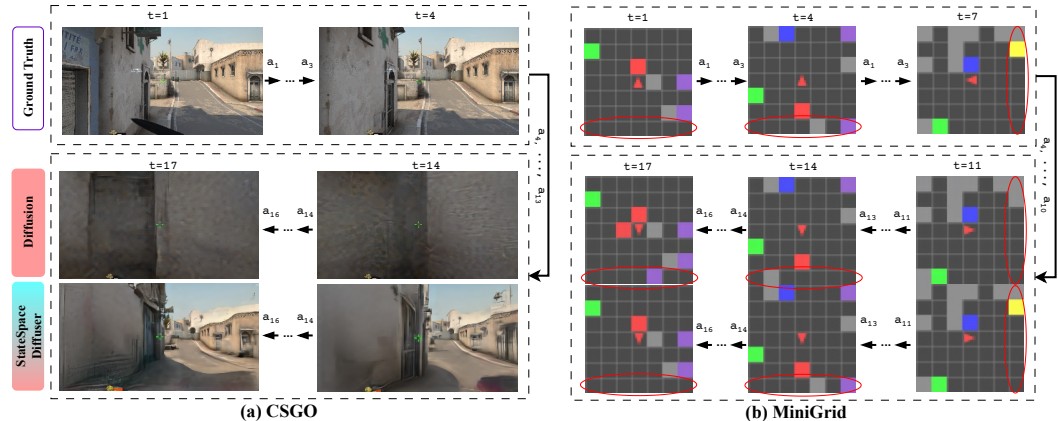

**Figure 6: Qualitative comparison of StateSpaceDiffuser and the baseline (DIAMOND).** Top row: input frames and actions. Bottom rows: generated frames under reversed actions. In CSGO - last 8 frames autoregressive prediction, in MiniGrid - next frame prediction.

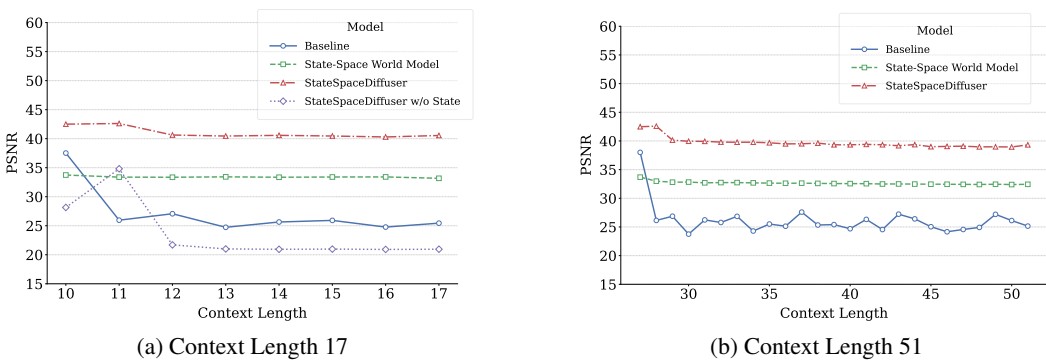

(a) Context Length 17                                          (b) Context Length 51

**Figure 7: Recall performance on MiniGrid for two context lengths.**

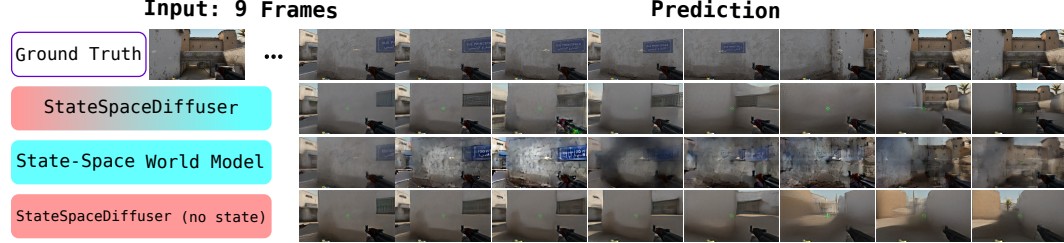

**Figure 8: CSGO Ablations.** We show that without the state-space features from the Long Context Branch, StateSpaceDiffuser loses its context preservation ability. We also show that while State-Space World Model demonstrates long-context memory, the produced images are of subpar visual quality.

**Forward-Backward Imagination Evaluation on CSGO.** Similarly to MiniGrid, we evaluated the recall abilities of our model on the CSGO dataset, a visually complex environment in a 3D world. In CSGO most actions are gradually executed over a sequence, and there is a compounding effect on content change (e.g. jump unfolds over many frames). For a high impact evaluation we decide to give only the first half of the sequence and continuously produce the second half (reverse) by feeding generated frames. As actions are motions at varying levels, the final frame may contain the correct content memorized but with low PSNR, as the camera position and scene geometry might be slightly shifted. Therefore, instead of fidelity metrics we perform a user study where the 12 participants judge whether images produced by StateSpaceDiffuser are closer in content to the ground truth compared to the diffusion baseline (details - in App. D.4). Our rating is in the range $[-1, 1]$, with 0 being borderline, -1 - preference toward the baseline, 1 - preference for StateSpaceDiffuser. The results shown in Fig. 5 demonstrate a clear preference of the users for StateSpaceDiffuser over the baseline

| Model | Avg. PSNR | Fin. PSNR | SSIM |
|---|---|---|---|
| *Context Length 100* | | | |
| DIAMOND | 26.39 | 26.24 | 0.95 |
| State-Space World Model | 31.65 | 30.89 | 0.96 |
| **StateSpaceDiffuser (Ours)** | **37.99** | **35.87** | **0.98** |

| Model | Avg. PSNR | Fin. PSNR | SSIM |
|---|---|---|---|
| *Context Length 150* | | | |
| DIAMOND | 24.35 | 24.20 | 0.94 |
| State-Space World Model | 27.93 | 26.98 | 0.94 |
| **StateSpaceDiffuser (Ours)** | **30.75** | **28.93** | **0.96** |

Table 2: **Generalization to Longer Context.** Our model, trained on context length 50, generalizes to longer sequences (context 100 and 150).

for both prediction in the 15th frame (rating **0.20**) and 17th (last) frame (rating **0.24**). Fig. 6 (a) shows a sample of CSGO imagination in different time steps, demonstrating that while the baseline fails to recall the correct content, the StateSpaceDiffuser correctly produces the details. (More in App. D.1)

**State Features Ablation.** We study the utility of the state-space features provided to the Generative Branch in our StateSpaceDiffuser model. We take a trained model and perform a MiniGrid evaluation by replacing the output features of the Long-Context Branch with zeros before passing them to the Generative Branch. In Tab. 1 we show that this causes the performance to quickly drop even below baseline performance, clearly demonstrating that the features are highly utilized. In Fig. 8 we demonstrate the same effect on CSGO. Without state features, the model hallucinates; without diffusion, the state-space model remembers but produces poor visual quality. (More in App. B.7), B.8)

**Generalization to Longer Context.** In this experiment we show that StateSpaceDiffuser operates on much longer contexts without finetuning. We evaluate our model trained on context length 50 on lengths 100 and 150 using a new MiniGrid test set with longer sequences. Tab. 2 shows that StateSpaceDiffuser successfully generalizes to longer context, keeping a significant gain over the baselines. Analogously, in App. B.3, we show generalization from context length 16 to length 50.

## 5.3 Strengths, Limitations and Scalability

Apart from the already established generalization across context length, via extra experiments, we find that StateSpaceDiffuser is able to generalize across visual complexity (App. B.6) and can recover from strong motion artifacts (App. D.2). Our model can recover from input noise in future steps, but is clearly affected by it on the current steps (App. B.5). Our lightweight StateSpaceDiffuser was trained under a fixed compute budget. The lightweight diffusion decoder (no large pretrained backbone) can yield visual artifacts in long rollouts. Replacing the decoder with a better, larger one, can improve visual sharpness without changing the method. Our lightweight single-layer Long-Context Branch compresses the context into a low-dimensional state (256), which can cause loss of detail in extended rollouts, especially in complex environments (App. D.2). Scaling the SSM (state dimension/heads/parameters/layers) is expected to reduce high-frequency decay over time. The separation in training enables separately scaling each branch before combining them.

## 6 Conclusion

We introduced **StateSpaceDiffuser**, a hybrid model that combines state-space representations with diffusion to enable long-horizon visual world modeling. By decoupling global context modeling (via a state-space backbone) from high-fidelity synthesis (via diffusion), our model retains global context over many steps at essentially no additional computational cost. The resulting representation alleviates the drift and inconsistency that plague conventional diffusion-only systems in long sequences.

Experiments on MiniGrid and CSGO validate our method's consistency and fidelity across long sequences. In the forward-backward protocol with horizon 50, StateSpaceDiffuser improves average PSNR by **51.9%** over the diffusion baseline and achieves a final-frame PSNR of **39.32** versus **25.14** for DIAMOND on a long context length of 50 frames. Human raters also favor our generations for long-context consistency (Fig. 5).

Our results establish state-space diffusion as a scalable and consistent solution for long-context visual generation. We believe that bridging state-space reasoning with diffusion generation is a promising direction for robust, long-horizon world modeling, and we hope this work lays a solid foundation for future research in temporally coherent visual prediction.

# 7 Acknowledgments

This research was partially funded by the Ministry of Education and Science of Bulgaria (support for INSAIT, part of the Bulgarian National Roadmap for Research Infrastructure). This project was supported with computational resources provided by Google Cloud Platform (GCP).

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

# Appendix

## A  Training Details

For MiniGrid, our DIAMOND baseline takes 30x30 images and upscales them to 144x144, for CSGO - 30x56, upscaled to 150x280. The Long-Context Branch contains the Cosmos tokenizer, which decodes in powers of two. Therefore, The Long Context Branch takes 32x32 for MiniGrid and 144x272 for CSGO. In training, the downscaling is done from the high-resolution image with bicubic interpolation (as in DIAMOND).

In our Long-Context Branch, the Cosmos Tokenizer tokens are flattened to produce features of size 1296 (MiniGrid) or 2448 (CSGO). Action dimensions in the Long Context Branch (and the State Space World Model) is 16 - they are concatenated to the visual features. We use a single Mamba layer with state size 256, the input is expanded 4 times with an MLP inside the Mamba layer, the internal convolution dimensions are 4.

We use 8 A100 GPUs for all models except for the MiniGrid State Space World Model, which was trained on 4 A100 GPUs for MiniGrid models. All models use the Adam optimizer. The State Space World Model is trained with a learning rate of $5e^{-5}$ and batch size 136 (MiniGrid) or batch size 272 (CSGO). Both the MiniGrid and CSGO models are trained for 70k iterations on sequence size 16. StateSpaceDiffuser includes 600M parameters, and is trained with a learning rate $1e^{-4}$, weight decay $1e^{-2}$, grad norm clip 10 and batch size 64. The MiniGrid model is trained for 77k iterations, CSGO - 220k iterations. For upscaling in MiniGrid - we train the sampler for 27k iterations after training the denoiser for predicting low-resolution next frame. In CSGO we achieved our best results with the upsampler, part of the weights originally provided by DIAMOND.

For all models, we loaded the weights of a pre-trained State-Space World model into StateSpaceDiffuser. For MiniGrid, while models trained on sequence length 16 performed well, in inference, we

| Backbone | Avg. PSNR↑ | Fin. PSNR↑ | SSIM↑ |
|---|---|---|---|
| *Context Length 16* | | | |
| DIAMOND | 27.13 | 25.44 | 0.95 |
| **Mamba** [31] | **34.41** | **37.81** | **0.97** |
| *Context Length 50* | | | |
| DIAMOND | 26.13 | 25.15 | **0.95** |
| **Mamba** [31] | **27.62** | **29.75** | 0.94 |

Table 3: **Performance Gains, Normalized By Computational Cost.** Strong gains are still observed, suggesting that the gains are much higher than the cost.

| Backbone | Avg. PSNR↑ | Fin. PSNR↑ | SSIM↑ |
|---|---|---|---|
| *Context Length 50* | | | |
| LSTM | 26.80 | 26.91 | 0.93 |
| GRU | 27.40 | 26.68 | 0.93 |
| **Mamba** [31] | **32.64** | **32.44** | **0.96** |

Table 4: **Evaluating Sequence Models.** The superior performance of Mamba leads to our choice of an SSM in our Long-Context Branch.

| Backbone | Avg. PSNR↑ | Fin. PSNR↑ | SSIM↑ |
|---|---|---|---|
| *Context Length 16* | | | |
| S4 [33] | 24.29 | 24.31 | 0.91 |
| **Mamba** [31] | **27.09** | **27.87** | **0.94** |

Table 5: **Evaluating State-Space Backbones.** The superior performance of Mamba leads to our choice to use this method as our Long-Context backbone.

achieved our best results for both context length 16 and 50 by using the model trained on context length 50. For CSGO, we used the State-Space World Model weights for context length 16 and also evaluated on context length 50.

When we evaluate DIAMOND and StateSpaceDiffuser, we use 5 denoising steps to denoise the next observation and 10 to upscale it.

## B    StateSpaceDiffuser Properties

### B.1    Performance/Cost Tradeoff

Our Long-Context Branch adds very little computation to the diffusion model, and in exchange it offers significant improvements in consistent generation. In inference with batch size of 1, we measure DIAMOND to require 909.515 GFLOPS (4 input frames). The Long Context Branch with context length 16, requires only 5.5 GFLOPS for context length 16 and 16.741 GFLOPS for context length 50. That is only 0.6% of all inference computations of the full model, for sequence size 16, and 1.8% for sequence size 50.

We show that the gains from the Long-Context Branch surpass its computational cost by a large margin. To account for the cost in our StateSpaceDiffuser scores, we normalize them by multiplying by $1 - 0.006$ for sequence size 16 and $1 - 0.018$ for sequence size 50. The normalized scores are reported on Tab. 3. They confirm that there is still a significant gain in performance despite the normalization.

### B.2    Long-Context Architecture Comparison

We compare the choice of the SSM in State-Space World Model with other popular sequence processing models - LSTM and GRU. We train on context size 50 with our MiniGrid setup. We add a linear layer on the input and output (dim 256, with ReLU activation) of the models. Tab. 4 shows that Mamba outperforms the other sequence models in long-range temporal dependencies, while remaining computationally efficient. This confirms the conclusion in the original paper [31].

In addition, we consider the choice of an SSM model itself. We consider the S4WM model the closest in spirit model in literature. However, as their model code is not available, comparing an S4 backbone to our choice of Mamba serves as the closest we can get to a comparison. We train State-Space

| Model | Avg. PSNR↑ | Fin. PSNR↑ | SSIM↑ |
|---|---|---|---|
| *Context Length 16* | | | |
| DIAMOND | 27.13 | 25.44 | 0.95 |
| State-Space World Model | 29.71 | 31.34 | 0.96 |
| **StateSpaceDiffuser (Ours)** | **34.62** | **38.04** | **0.98** |
| *Context Length 50* | | | |
| DIAMOND | 26.13 | 25.15 | 0.95 |
| State-Space World Model | 27.25 | 24.49 | 0.93 |
| **StateSpaceDiffuser (Ours)** | **28.12** | **30.30** | **0.96** |

Table 6: **MiniGrid Quantitative Evaluation of Long Context Awareness, Trained on Context Length 16.** Our models generalize their performance from a smaller to a longer sequence.

| Noise | 27 | 28 | 29 | 30 | 31 | 32 | 33 | 34 | 35 | 36 | 37 | 38 | 39 |
|---|---|---|---|---|---|---|---|---|---|---|---|---|---|
| SSM | 15.22 | 16.62 | 15.36 | 15.18 | 14.85 | 16.55 | 23.44 | 26.60 | 28.42 | 29.35 | 29.75 | 29.75 | 30.00 |
| Full | 15.50 | 16.90 | 15.49 | 15.30 | 14.94 | 16.65 | 23.28 | 26.51 | 28.35 | 29.33 | 29.75 | 29.95 | 29.85 |

| Noise | 40 | 41 | 42 | 43 | 44 | 45 | 46 | 47 | 48 | 49 | 50 | 51 |
|---|---|---|---|---|---|---|---|---|---|---|---|---|
| SSM | 29.93 | 30.06 | 29.99 | 30.12 | 30.14 | 30.45 | 30.11 | 30.18 | 30.19 | 30.08 | 30.19 | 30.19 |
| Full | 30.09 | 29.88 | 30.11 | 30.08 | 30.41 | 30.12 | 30.13 | 30.17 | 30.12 | 30.20 | 30.13 | 30.15 |

Table 7: **Noise robustness across steps.** Our method is affected on the noisy frames but quickly recovers in further steps. **SSM** denotes the noise is added only to the Long-Context Branch; **Full** denotes the noise is added on the Generative Branch as well.

World Model with S4 and Mamba on MiniGrid using a context length of 16 and comparing their performance. We keep the context small to account for the larger amount of training iterations usually needed by SSMs. As seen on Tab. 5, Mamba outperforms S4, achieving a 9.1 PSNR improvement on average. As Mamba introduces a dependency on the input of the state update, this clearly benefits its ability to perform in long context.

### B.3 Generalization Across Context Length

In Tab. 6, we show the evaluation results of models trained in context size 16, on both context length 16 and 50. A reasonably good performance demonstrates that the models do not overfit on a particular sequence size and still perform well in a context length longer than it has been trained with. While the State-Space World Model exhibits uncertainty in its predictions for a longer context without training (blurriness, color deviations), its features prove useful for StateSpaceDiffuser, with the Generative Branch producing a higher-fidelity result. StateSpaceDiffuser noticeably outperforms the baseline on context length 50.

### B.4 Stability Across Seeds

To show the stability of our results, we perform evaluation of our MiniGrid model, on context length 16, under 4 different seeds. We obtain $41.00 \pm 0.008$ Avg. PSNR, $40.52 \pm 0.019$ Fin. PSNR, $0.98 \pm 0.0004$ SSIM. We also perform evaluation over 4 seeds of a larger-scale, more expensive evaluation - our new 100 context length experiment from Sect. B.3. This results in: $37.99 \pm 0.002$ Avg. PSNR, $35.88 \pm 0.029$ Fin. PSNR, $0.98 \pm 0.000004$ SSIM. In both cases, the obtained metrics are extremely stable and consistent between seeds, with a low standard deviation.

### B.5 Robustness to Noise

We test the robustness of our method by adding noise in the middle section of the rollouts that serve as context. In specific, we consider context length 50 in MiniGrid and add Gaussian noise (std 2.5) to the 11 frames in the middle. We consider two cases: 1) adding noise to the SSM input only; 2) adding noise both to the SSM input and the diffusion model input. Results are shown on Tab. 7 at different steps of prediction after the middle frame. We observe that for the specific frames with added noise, the performance decreases. On those frames content can disappear and context is not correctly recalled. However, in both cases, within 4 steps after the noisy frames (after frame 34 - 5

| Model | Avg. PSNR↑ | Fin. PSNR↑ | SSIM↑ |
|---|---|---|---|
| *Low Complexity* | | | |
| Baseline (low complexity) | 26.09 | 25.60 | 0.95 |
| **Ours (low complexity)** | **36.72** | **35.78** | **0.97** |
| *Middle Complexity* | | | |
| Baseline (middle complexity) | 27.27 | 26.70 | 0.94 |
| **StateSpaceDiffuser (Ours))** | **39.68** | **39.32** | **0.98** |
| *High Complexity* | | | |
| Baseline (high complexity) | 23.09 | 22.87 | 0.93 |
| **StateSpaceDiffuser (Ours)** | **31.67** | **30.87** | **0.97** |

Table 8: **Generalization Across Visual Complexity.** Our approach is shown to consistently outperform the baseline on multiple levels of visual complexity without finetuning.

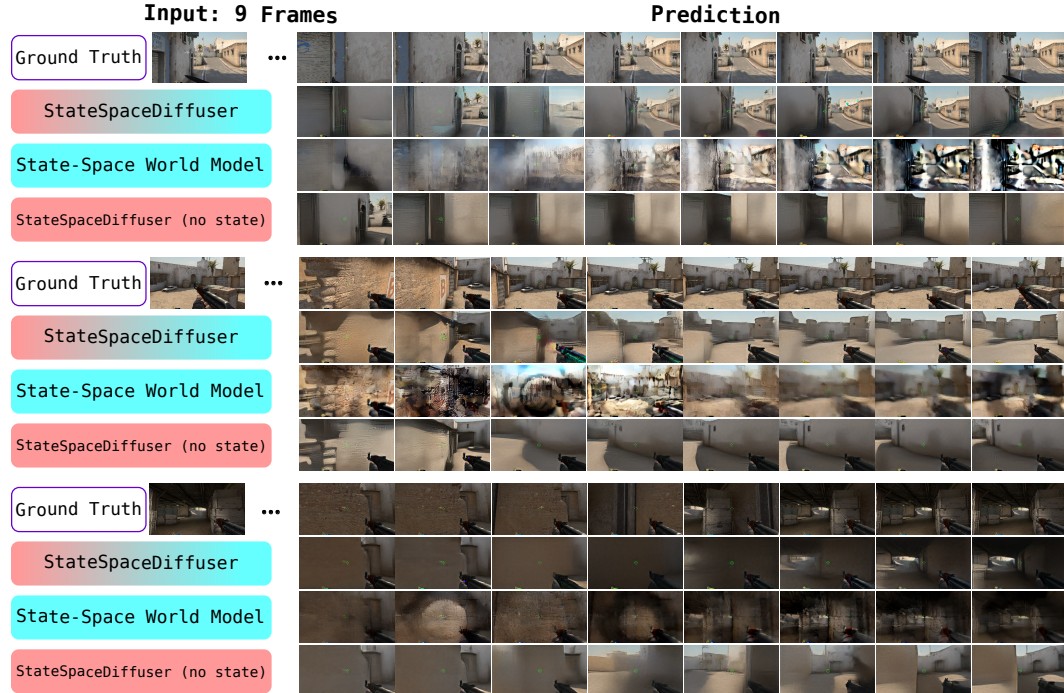

Figure 9: **CSGO Ablations.** We show that without the state-space features from the Long Context Branch, StateSpaceDiffuser loses its context preservation ability. We also show that while State-Space World Model demonstrates long-context memory, the produced images are of subpar visual quality.

noisy frames + window size 4), the memory and content recovers and is correctly predicted, with stable performance until the last frame. The lower scores suggest some loss in performance. However, the fact that memory recovers after the noise suggests a certain level of robustness to noise.

## B.6  Generalization Across Visual Complexity

In this work, we have evaluated on 3 different environment setups with increasing level of complexity - the very constrained Simple MiniGrid, free navigation in a maze (MiniGrid), and a 3D first-person environment (CSGO). To further compare the performance of StateSpaceDiffuser across environments with different visual complexities, we generate 2 more variants of our MiniGrid dataset based on visual complexity. We define complexity as number of markers and complexity of the maze walls (values in the range $[1, 5]$). We generate a dataset with low complexity (200 markers, difficulty 3), and with high complexity (450 markers, difficulty 5). Our original MiniGrid dataset lies in between in terms of complexity (360 markers, difficulty 4). We take our StateSpaceDiffuser pretrained model on context length 50 (middle complexity) and evaluate it on the new datasets with varying difficulties (without any finetuning).

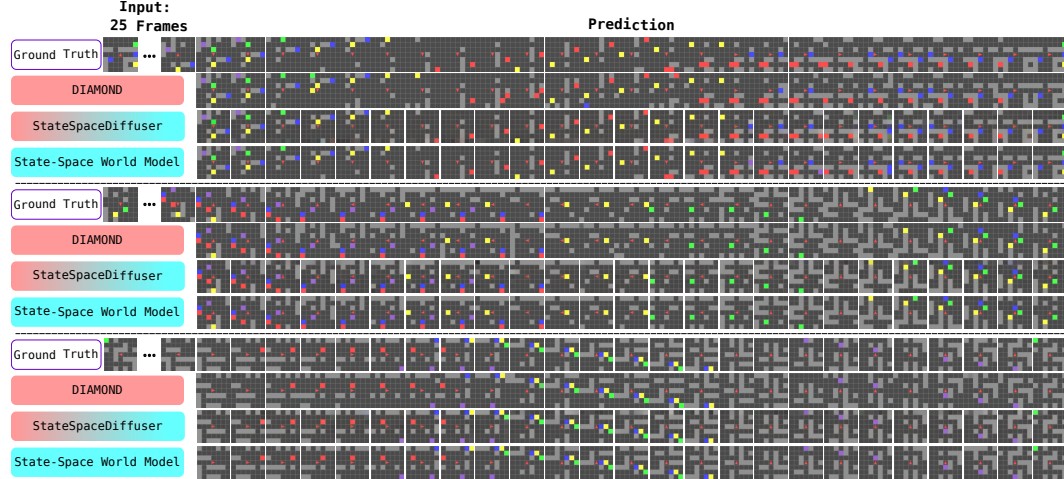

Figure 10: **Qualitative Results with Context Length 50 on MiniGrid.** StateSpaceDiffuser demonstrates long-context preservation compared to DIAMOND and better visual fidelity compared to State-Space World Model.

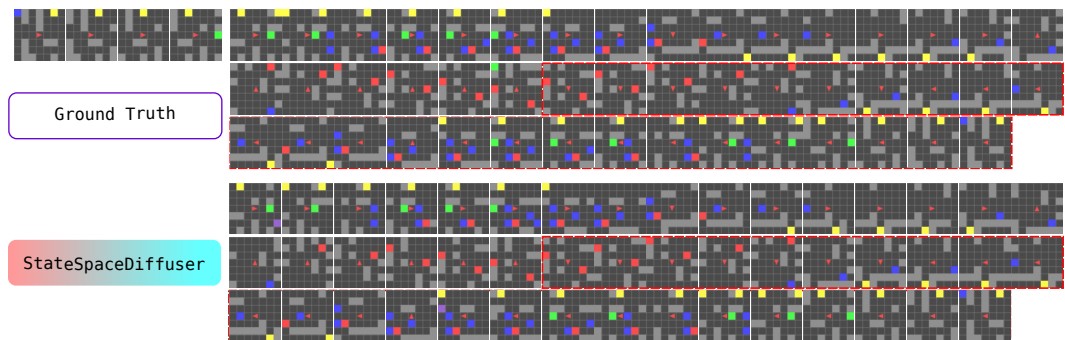

Figure 11: **Full Sequence Sample with Context Length 50 on MiniGrid.** Demonstrates a full 50-frame context and the per-frame predictions. In red are shown the frames from the second half - returning to the initial position.

The results in Tab. 8 suggest that our model is better able to generalize to lower complexity rather than higher complexity. However, in all cases, the performance remains higher than the baseline (DIAMOND).

### B.7 State Features Ablation

To expand on the state ablation experiment results from Sect. 5, in Fig. 9 are shown more examples comparing StateSpaceDiffuser with and without a state. The results reconfirm that StateSpaceDiffuser loses its special ability to memorize long context after we zero out the states that were produced before passing them to the Generative Branch. Without the state, the model can no longer recall long-context information and hallucinates new locations.

### B.8 Comparison with State-Space World Model

In Fig. 9 are shown extra examples of the State-Space World Model (used as a Long Context Branch in StateSpaceDiffuser) versus the result of StateSpaceDiffuser. It is clearly seen that the State Space World Model produces more artifacts and more visually unappealing images. However, its important property to reconstruct previously observed content leads to its features being crucial for StateSpaceDiffuser to exploit as part of its Long Context Branch. In the shown examples, the images are predicted in an autoregressive manner by re-encoding StateSpaceDiffuser's output from the previous steps. We have observed that this approach is much more stable than feeding the prediction

of State-Space World Model to itself. The latter produces very blurry images with no discernible detail. Note that the State-Space World Model has not specifically been trained with an autoregressive approach. This confirms that in StateSpaceDiffuser, the diffusion component stabilizes the state-space component, which in turn improves the long-context abilities of the diffusion component.

Compared to MiniGrid, here we see much more pronounced artifacts on the results of State-Space World Models, which are then cleaned up by the Generative Branch in StateSpaceDiffuser. This confirms the potential of StateSpaceDiffuser particularly for complex environments.

## C  MiniGrid Evaluations

In this section, we expand our discussion on our model's performance on the simpler MiniGrid dataset and offer additional qualitative and quantitative results.

### C.1  MiniGrid Qualitative Evaluation

In Fig. 12 and Fig.10 we show visual samples with the forward-backward evaluation used in our quantitative results. We assess the StateSpaceDiffuser model, trained on sequence length 51 and presented in the main paper and its corresponding State-Space World Model.

Note that in our standard evaluation on MiniGrid, at each time step we give the ground truth sequence up to that step and predict the next observation. In a single time step the agent takes a fixed motion in one of four directions and reveals exactly 1 row or column of the environment depending on the direction. As most of the content of the next frame is present in the previous frame, the unknown content is only contained within the new revealed area. In the first half of the sequence, the agent explores by navigating the maze. In Fig. 11 we show one such traversal and the predicted next frame for each time step from StateSpaceDiffuser. It is observed that the new revealed area is predicted far from the ground truth in the first half of the sequence. This is expected as this area has not yet been observed. However, in the second half, the return along the path, the ability to recall the content from the long given sequence helps to predict the correct content of the repeatedly revealed areas. Therefore, in all our evaluations we have made the decision to only consider the prediction quality of the second half of the second half of the sequence.

In context length 16 - Fig. 12, we clearly observe poor performance of the diffusion-only baseline, DIAMOND, as this model has no method to take into account long context. Looking closely at the State-Space World Model's output, we observe shifts in color and inconfidence in the content of the new revealed areas. The effect is subtle and varies in the sequence, but it is noticeable. This effect causes a drop in fidelity metrics and is a direct consequence of the non-variational approach of predicting the next frame from State-Space World Model. In contrast, StateSpaceDiffuser is free of such artifacts and predicts closer to the ground-truth images. Still, as it is conditioned on the state-space features, it can be affected by significant uncertainty in the content of particular grid cells.

The observations are even more pronounced for context length 51 - Fig. 10. Later in the sequence, the State-Space World model tends to increase its shifts from the ground truth (dar squares appear darker, grey squares tend to fade). While on a simpler dataset like MiniGrid such artifacts are less noticeable, for a more complex setup like CSGO this becomes more apparent.

### C.2  Imagination Qualitative Results

Additionally, instead of giving the ground truth sequence at every step, we also attempt to give only the first half and feed already predicted frames for time steps in the second half of the sequence (imagination). In this way the ground-truth frames in the second half are never seen by the model (same as the setup we have in the CSGO dataset). This is a more challenging setup, in which the content cannot be copied from the previous ground truth frame, and any errors in the current frame prediction propagate into the next frame.

We show qualitative results in imagination in Fig. 13. It is observed that in this more complex setup the diffusion baseline quickly drifts away from the context, and the entire image no longer corresponds to past context. In contrast, because of the incorporation of state-space features, the StateSpaceDiffuser is noticeably better at preserving the content of previous steps.

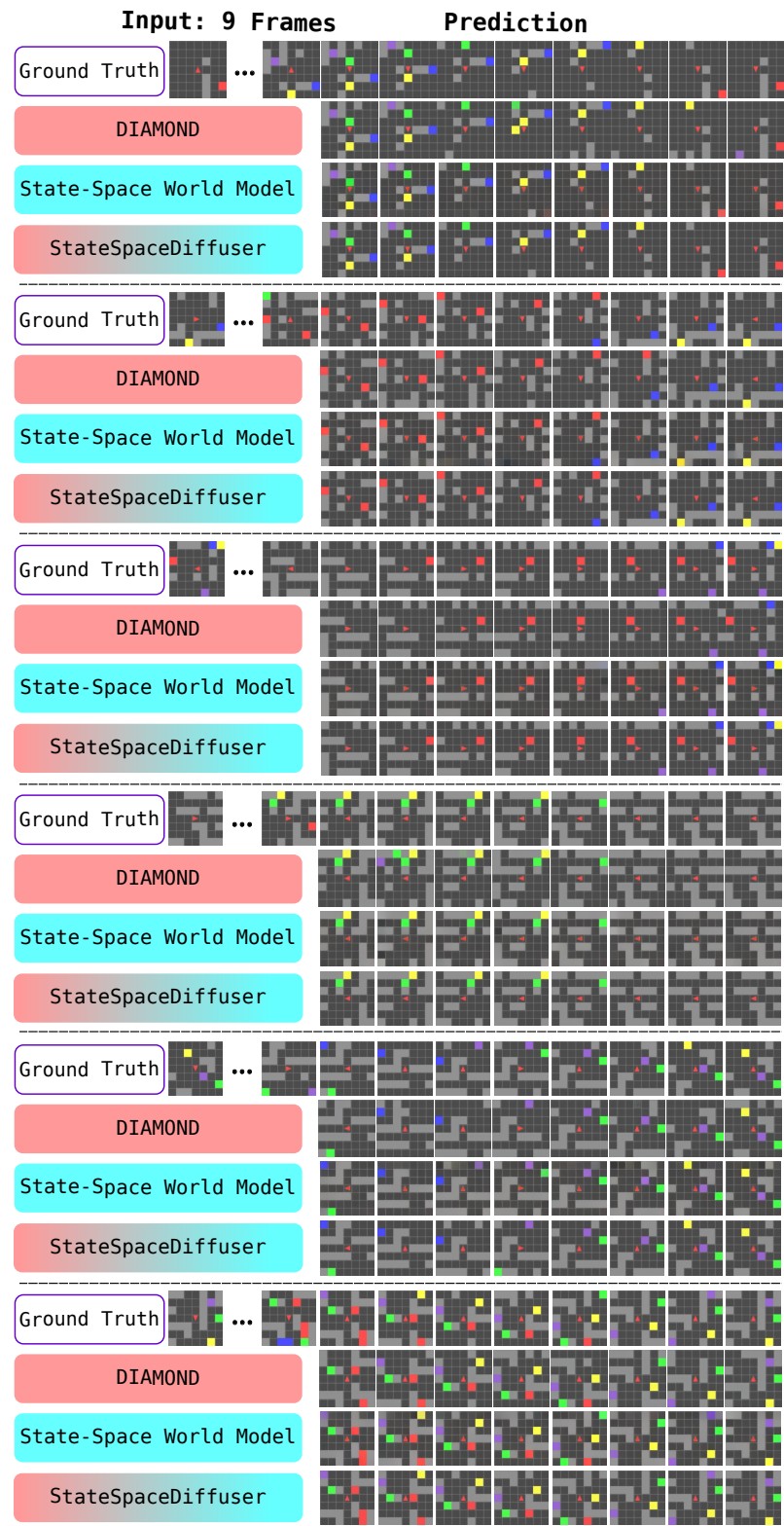

Figure 12: **Qualitative Results on MiniGrid.** Compared to DIAMOND, State-Space World Model is able to recall past content better but lacks in certainty and visual fidelity. However, StateSpaceDiffuser is able to both to consider long context and to produce a high quality image.

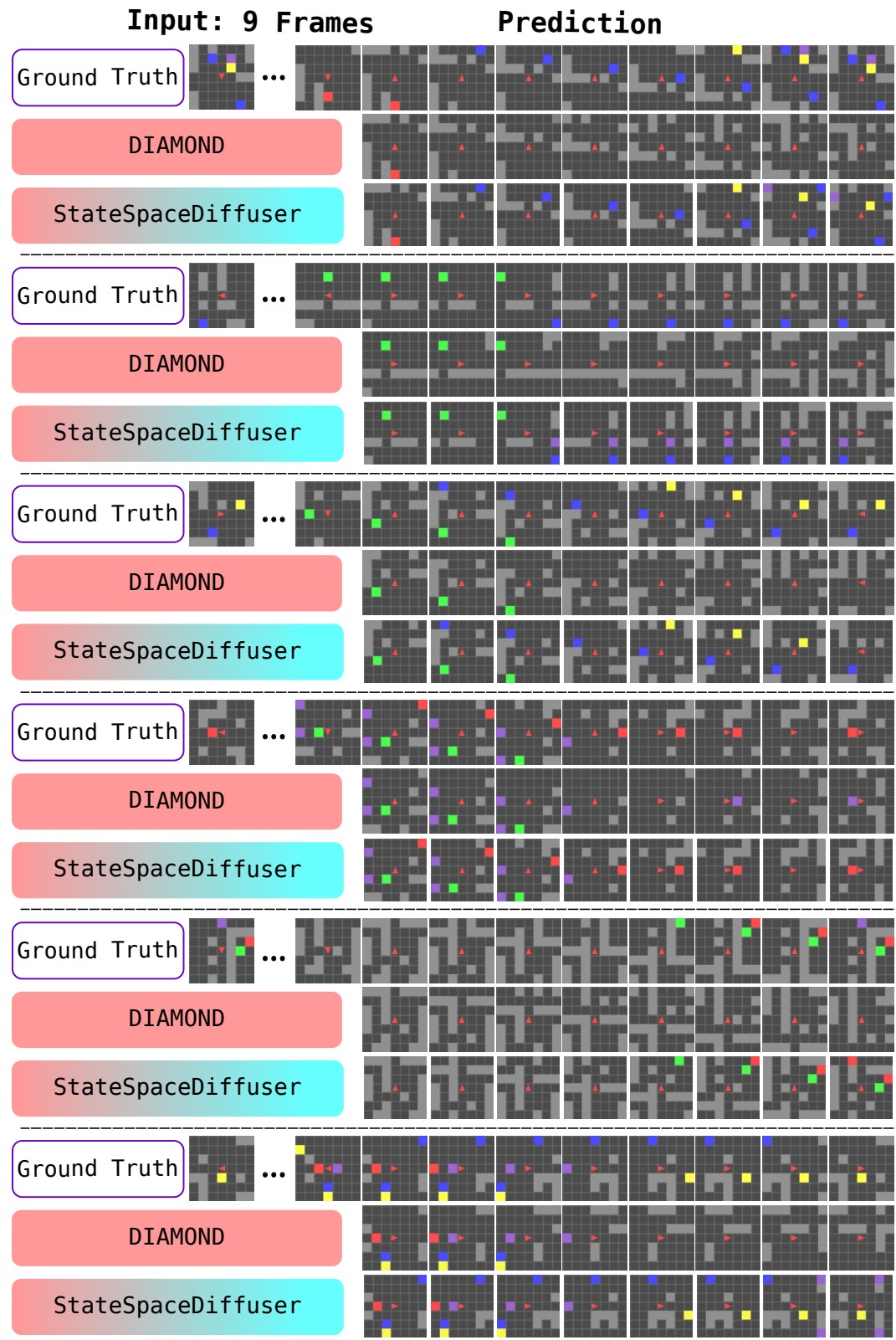

Figure 13: **Imagination Qualitative Results on MiniGrid.** Frames are consecutively generated given previously generated frames. StateSpaceDiffuser shows clear superiority on context preservation.

# D  CSGO Evaluations

## D.1  CSGO Qualitative Evaluation

We show additional qualitative examples in Fig. 14. We show the last 8 frames (predicted without the use of the last 8 ground-truth frames), given 9 frames of the mirrored version of the sequence. The last predicted frame uses context length 16. In the results, it is observed that DIAMOND - our diffusion baseline - is unable to recall a previous context that was left beyond the 4 input frames it accepts as input per step. In contrast, StateSpaceDiffuser is able to predict the correct content through its Long Context Branch.

In addition, we evaluated our model (trained on context length 16) in a longer context of 50 frames. In Fig. 15 we show examples, where the last 8 frames are predicted in an autoregressive manner, while the first 43 frames are given as ground truth (their corresponding predicted frames are also depicted). Therefore, the context for the last frame is of 43 ground-truth images and 7 generated images. We show the last 26 frames in the sequence, as the first 25 are a mirrored version of them. In this challenging setting, we observe more artifacts and significantly less memory capabilities. However, the model is capable of recalling some visual cues on this context length that were visible

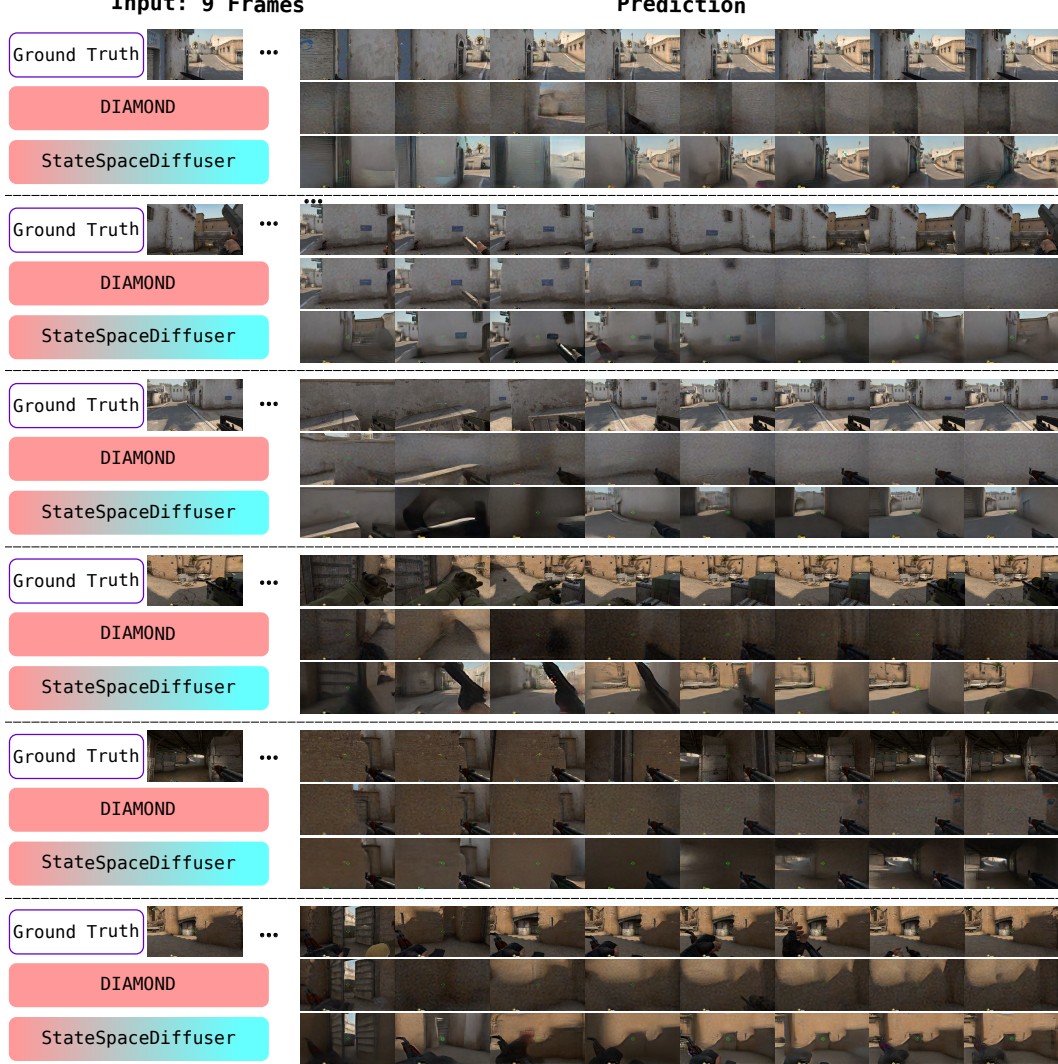

Figure 14: **Our StateSpaceDiffuser model is able to recover from insufficient information in the short context**, while the diffusion baseline - DIAMOND, has no mechanism to do so.

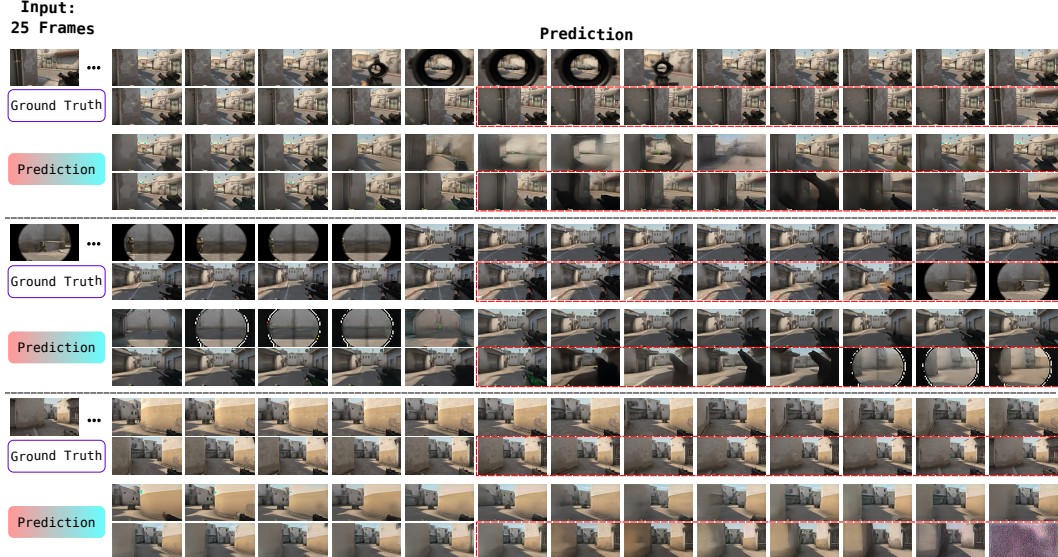

Figure 15: **StateSpaceDiffuser on Context Length 50 on CSGO.** In red are marked the imagined frames and the corresponding ground truths. The model is able to reconstruct frames seen at the start of the sequence.

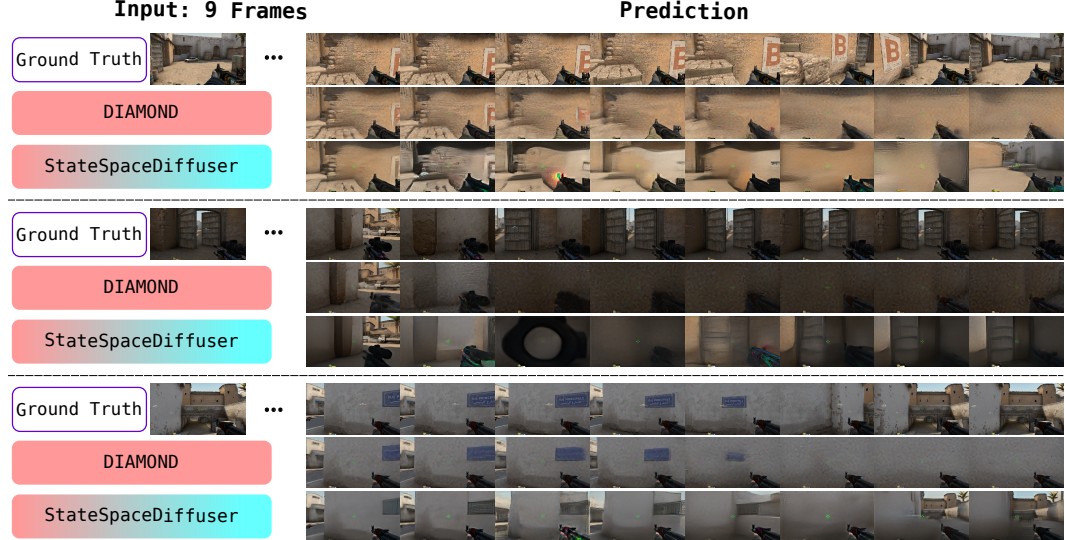

Figure 16: **Recovery from a strong action.** Strong actions cause significant artifacts in DIAMOND. While this behavior is inherited by StateSpaceDiffuser, in contrast, it quickly recovers from the artifacts using the state-space features.

only at the start of the sequence, as visible on the examples - the red roof, looking through the scope, revealing a door.

## D.2 Strengths and Limitations of StateSpaceDiffuser

DIAMOND has a limitation, which causes significant artifacts when the action is strong and results in a larger visual change (e.g., large turn). Our Generative Branch is based on DIAMOND and hence has a similar limitation. However, while DIAMOND's artifacts tend to affect the entire predicted sequence, the StateSpaceDiffuser has the property to recover in subsequent steps by making use of the state-space representation to recover the content. This is visible throughout Fig. 14, but also particularly in the examples shown in Fig. 16.

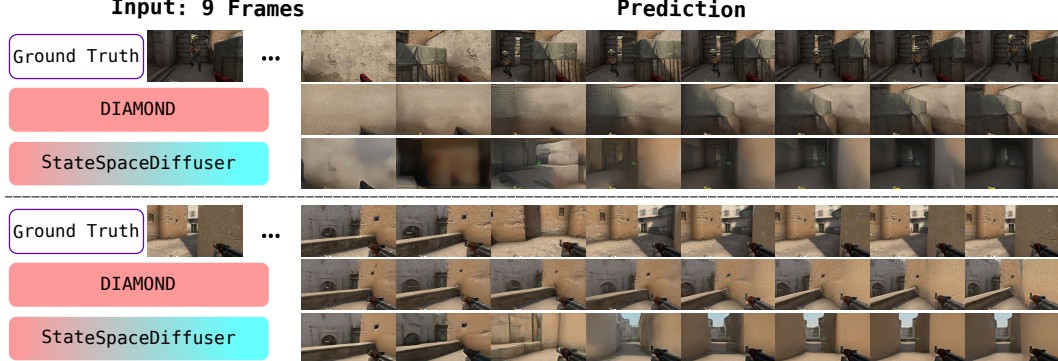

Figure 17: **Limitations.** Our method sometimes only reconstructs coarse features and leaves details out. Even in these cases, the content appears closer to the ground truth than the diffusion baseline.

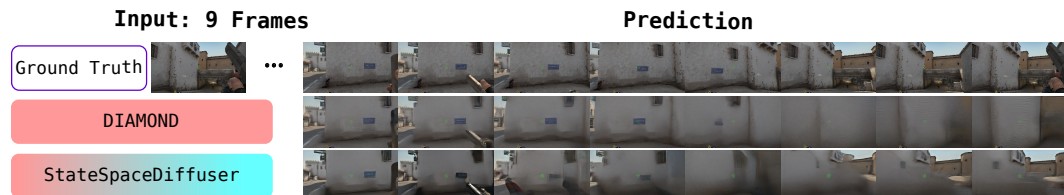

Figure 18: **Measuring Content Preservation.** On the shown sample, visually our method preserves the content better than the baseline. However, the PSNR fidelity metrics do not match this conclusion. We conclude that fidelity metrics do not match well the goal of content preservation.

The effect of the strong action is one of our main motivations to use autoregressive-style prediction for the CSGO models rather than a single frame prediction given a full context (as done with MiniGrid). While MiniGrid has actions with constant measurable visual effect, CSGO's actions vary in strength. We observe that many sequences consist of almost no motion, divided by a few strong actions. In the single-action prediction task, weak motion would not require much new content to be generated. This often results in a copy of the previous frame, and not much improvement is expected compared to the baseline. Conversely, with strong actions, the baseline and StateSpaceDiffuser exhibit inconfidence in the next frame. This affects the short context window, filling it with artifacts. In comparison to the baseline, the StateSpaceDiffuser is able to recover in the following frames.

While our method's content is consistently closer to the ground truth than the diffusion baseline, it sometimes is only able to preserve coarse features (colors, general shape of the scene) and is less effective with finer details. We show this in Fig. 17. It is visible that sometimes our method misses showing an object (crate in the first example) or just preserves the general shape of a scene (second example). Even with those limitations, the model often outperforms the baseline. As a higher level of detail requires larger memory capacity, we believe that with more computational resources, scaling the Long-Context Branch can aid these issues.

### D.3 Quantitative vs. Perceptual Consistency

As previously discussed, in contrast to MiniGrid, the complexity of the CSGO environment makes fidelity metrics such as PSNR unsuitable for evaluating content consistency. CSGO is characterized by actions with variable motion unfolding over multiple frames. Long rollouts accumulate small motion mismatches into camera-view drift, so later frames need not match ground-truth pixels. Content often remains perceptually similar while viewpoint and details differ, and PSNR under-reports this similarity. We demonstrate this by computing the fidelity metrics of an example. For the baseline, we obtain **20.77** Avg. PSNR and **16.17** Fin. PSNR. For StateSpaceDiffuser - **19.36** Avg. PSNR and **16.11** Fin. PSNR. Given the metrics, in this example, our model does not differ significantly from the baseline in terms of quality of the last frame and is somewhat worse on average. However, when looking at the visual results in Fig. 18, we clearly see more similarity to the ground truth in the content of our method than in the baseline. The viewpoint, details, object proportions, and parts of

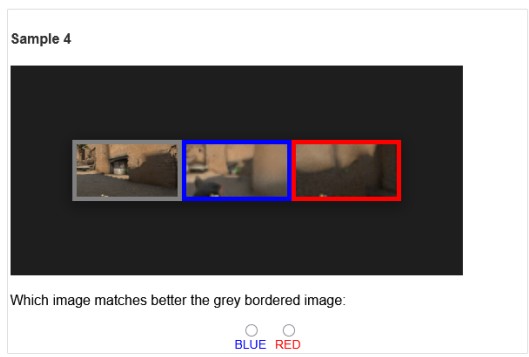

Figure 19: **User Study Sample Question.**

the scene do not match exactly, causing the metric to be worse. Due to this inadequacy, and following common practice, we opt for a user study to evaluate the quality of the CSGO results.

### D.4 User Study Details

The user study is performed by first selecting 40 examples where long-context memory would be important - for example, content at the beginning of a sequence is covered up by a wall. The examples are picked from the ground truth images in the test set; the predicted images are not observed when selecting, in order not to bias the process. After generating the predictions from the diffusion baseline and StateSpaceDiffuser, we build two triples of images per prediction - for prediction horizons 15 and 17. This results in 80 total triplets. For each of them, we ask 12 participants to determine if the baseline or our model is better. The participants are coleagues and students from our institute, external to this project. In order to avoid bias, we shuffle the order of the baseline and StateSpaceDiffuser results for each sample, marking the first blue and the second red. The user is asked to compare the match of blue and red images to the ground truth image. An example question is shown in Fig. 19.

The text that the users saw is visible below:

*Thank you for taking part in our user study! For each question, you will see a row of 3 frames:*

- *First frame (grey border) - our ground truth*
- *Second frame (blue border) - a frame that attempts to resemble the first frame*
- *Third frame (red border) - a frame that attempts to resemble the first frame*

*Your task is to judge if the red or blue frame resembles more closely the grey frame. Rate each pair by selecting: Blue - Blue frame resembles the grey frame better; Red - Red frame resembles the grey frame better. Please base your decision on both the content and not the visual quality of the images.*

