# OpenReview forum: "StateSpaceDiffuser: Bringing Long Context to Diffusion World Models"
_NeurIPS.cc/2025/Conference — NeurIPS 2025 poster_

### Official Review · Reviewer_1SP4 · 2025-07-01

**Clarity:** 2
**Significance:** 2
**Originality:** 3
**Rating:** 5
**Confidence:** 4

**Summary:**

This paper introduces a method for incorporating long-context understanding into diffusion-based world models by combining a state-space model (SSM) with a diffusion model. The SSM processes long input sequences and maintains a latent environment state that accumulates information over time. This environment state, along with a short history of image frames, is then fed into the diffusion model to predict future world states in image space.

The key insight is that while diffusion models excel at generating high-quality images, they are limited to short sequences due to memory constraints. In contrast, SSMs can handle long sequences efficiently by recursively updating a compact internal state. The proposed approach leverages both strengths: the SSM captures long-term context, and the diffusion model handles high-fidelity image prediction.

To assess the model's long-term reasoning capabilities, the authors introduce an evaluation framework where the task is to observe a long sequence and then predict it in reverse. Experiments are conducted on two environments: a 2D maze setting (MiniGrid) and a 3D first-person shooter game.

**Questions:**

1. If the full sequence of 50 frames can be held in memory, is the problem essentially solved with a standard diffusion model? Is the real limitation just memory? It would be helpful if the authors discussed this more explicitly, as it seems central to the motivation for introducing the state-space model.
2. Clarification on MiniGrid Setup (L238): The paper states, "On MiniGrid, we generate 1 frame in the future..." Why is only one frame predicted here, whereas in CSGO the model generates the second half of the sequence? Section 3 does not clearly explain this discrepancy.
3. In (L247) the paper says, "We train and test on the same set of 34 samples, allowing for overfitting." Could the authors elaborate on the reasoning behind this choice? Is the goal to ensure that the state-space model is well-trained before evaluating the generative model and that is only possible on a small dataset? If so, how does this affect scaling to larger datasets?
4. The authors note in the supplementary that the state-space model shows more artifacts in CSGO than in MiniGrid, suggesting room for improvement. However, since the SSM must be trained separately, how scalable is this method to more complex real-world environments? 5. Would the quality of the generative model always be bottlenecked by the expressiveness of the SSM? The paper states, "We load the pre-trained weights and train the Generative Branch to obtain the final model." Is the SSM frozen during this stage, or is it fine-tuned jointly with the generative model? Additional details on the training pipeline would be very helpful.

**Ethical Concerns:**

["NO or VERY MINOR ethics concerns only"]

**Final Justification:**

Thanks to the authors for being active in the discussion phase. They addressed all my concerns.

I agree with Reviewer VVHr that some of the other reviews may be overly harsh in insisting on evaluations on more complex datasets as a requirement for acceptance. As long as the paper is well positioned with respect to concurrent and prior work, I think it is reasonable for the evaluation to be limited given the inevitable resource constraints, since we cannot expect a single paper to solve every problem. The authors have said they will include these discussions, which I believe is more than sufficient for acceptance.

I also think the way SSMs and diffusion models have been combined in this paper is non-trivial. After discussing with the authors, the approach seems well-motivated; it simply needs to be presented more clearly in the methods section, which the authors have agreed to do. So I would recommend the paper for acceptance.

**Limitations:**

A discussion on how the model would scale to larger datasets and failure models on CSGO were not sufficiently discussed. More details are mentioned in the Weaknesses section.

**Paper Formatting Concerns:**

Nothing in particular.

**Quality:**

2

**Strengths And Weaknesses:**

**Strengths**

1. The paper addresses an important problem: how to build long-term context understanding and how to build in a memory mechanism into generative world models. As the authors rightly point out, this capability is essential for a range of downstream tasks, and current models are limited to only operating on short time scales.
2. The evaluation task is intuitive and well-suited for assessing long-context reasoning. Predicting an episode in reverse after observing a few frames resembles a natural form of episodic recall.
3. The proposed model outperforms a diffusion-only baseline, demonstrating the effectiveness of combining a state-space model with a diffusion model.
4. The experiments are conducted in both a simple 2D environment (MiniGrid) and a challenging 3D first-person environment, providing a reasonable spectrum of test settings.

**Weaknesses**

1. While the paper tackles an important problem, the experimental settings feel somewhat limited. Both environments—MiniGrid and CSGO—are relatively constrained compared to real-world scenarios. This raises questions about the method's scalability and general applicability.
2. The state-space model (SSM) operates on a compressed state representation and this likely contributes to the poor reconstruction quality observed in Figure 6. I think this calls into question the practical utility of the method, or atleast makes the method not super impressive. The authors acknowledge the PSNR metrics limitations in the supplementary material, but I strongly feel such discussion belongs in the main text to properly contextualize the model’s performance.
3. Although the model outperforms a diffusion-only baseline, this result is not very surprising, given that the SSM has access to long-term history. The improvement itself doesn’t offer particularly surprising insights. In some sense it’s just a natural consequence of the architecture so it’s hard to say that it’s a major conceptual breakthrough.
4. The paper claims strong results but lacks a detailed discussion of failure modes. It’s unclear how often or why the SSM fails, especially given its lossy state representation. Without this, it's difficult to understand the model's limitations or how future work might build on it.
5. The need to train the SSM and diffusion model separately is briefly mentioned, but not adequately justified or explored. This separation could be a significant limitation, as it prevents end-to-end optimization and ties overall model performance to effectiveness of the SSM. It’s unclear whether the diffusion model is truly doing anything more than rendering from the SSM’s output.
6. The inclusion of a user study feels unnecessary and potentially misleading. Visual artifacts in Figure 6 already make it clear that the baseline underperforms, and qualitative metrics like PSNR or LPIPS would have provided more objective insight. A user study comparison seems uninformative in this context and could be omitted.
7. Writing quality could be improved, especially in making architectural choices and limitations more transparent. There are also quite a few typos and presentation issues which I discuss below.

**Writing and Clarity Suggestions**

*Introduction:*

1. The authors make the statement that Diffusion models are most often directly limited by their input window size. This might need more clarification --- Is this limitation due to memory constraints, or are current architectures fundamentally incapable of working with longer contexts?
2. A positioning relative to relevant recent work, such as Long Context Tuning for Video Generation (Guo et. al), which challenges the idea that diffusion models are strictly short-context could be useful.
3. The paragraph at the end that discusses metrics and points to Section 5 disrupts the flow.
4. Line 66: The statement; "effectively brings a state-space model and diffusion" is awkward.

*Data:*

1. A visual illustration of the task environment or gameplay (especially for the CSGO setup) would make it easier to understand what the model is actually trying to predict.

*Typographical and Caption Issues:*

1. Line166: Check for a possible typo
2. Figure 6: Do downward arrows indicate that both frames and actions are provided, or just actions? Clarify this in either the figure or its caption.
3. Figure 5 caption: Missing space after the title—should follow standard formatting

---

> ### Author Rebuttal · Authors · 2025-07-31
>
> We thank reviewer for the feedback and the comments about importance of our topic, the suitability of our evaluation framework, the effectiveness of our approach and the diversity of our test settings. We address the reviewer's comments below.
>
> **Computational Limitations of Existing Methods.**
> Even if 50 frames fit in memory, the real barrier is computational complexity. During training, our SSM updates a fixed-size state once per frame, so compute and activations grow linearly with sequence length $T$. A Transformer-based diffusion must still form full self-attention, which is quadratic $O(T^{2})$ in compute and memory. At inference the gap remains: the SSM’s per-step cost is constant, whereas even KV-cached attention or large-receptive-field CNNs scale linearly with the window. In other words, the SSM can process an increasing sequence in a streaming manner with constant latency. Thus a 50-frame cache alleviates memory but not the quadratic compute bottleneck, and long contexts are typically truncated, forfeiting long-range information.
>
> Long Context Tuning for Video Generation (Guo et al.)  trains with full attention over an expanded window (O(N²)), and at inference the per-step cost still scales with the window despite KV cache. We instead compress history into a fixed-size SSM state, enabling linear complexity and constant latency (even with an LCT decoder). Moreover, LCT’s asynchronous shot generation is ill-suited for interactive environments with per-frame actions.
>
> Furthermore, in contrast to methods which finetune/train new models to extend the context, we show that our method is able to extend to much longer sequences than trained without finetuning. We refer to Sup.Mat. Sect. 2.1 where we generalize from context size 16 to context size 50, and an experiment that can be seen in the response of Reviewer dCL402, where we generalize from context size 50 to up to 150.
>
> **SSM/Diffusion Training Separation.** We confirm that our training consists of two stages: 1) Training the SSM (State Space World Model) 2) Load the weights of the SSM, freeze them, and train the Generative Branch of StateSpaceDiffuser.
>
> Joint (end-to-end) training - from scratch or finetuned - underperformed. Training separately worked best. The two branches learn on different time scales: the SSM needs many sequence-level iterations to stabilize long-horizon state, while the diffusion model quickly fits a local denoising loss and ignores a still-adapting SSM. Finetuning didn’t help: diffusion gradients disrupt the SSM and the decoder reverts to immediate image cues. Freezing the SSM prevents drift and lets the decoder use long-term context. Importantly, the diffusion model does not just "render the SSM’s output": in Sup.Mat. Sec. 3.6 Fig. 11, an ablation shows the SSM alone produces low-quality frames with the right content, while StateSpaceDiffuser—given the same SSM features - produces visibly higher-quality, context-aware results, not a copy of the SSM output. Thus, the diffusion model isn’t bottlenecked by SSM visual quality.
>
> The reviewer is concerned if performing the training separately could be a scalability issue.
> We view the modular training as a scalability advantage, not a limitation. It decouples learning an efficient, fixed-size memory for long-range semantics (the SSM) from high-fidelity rendering (the diffusion decoder). This lets us upgrade either module without retraining the other (e.g., drop-in a stronger decoder, or train an SSM on longer sequences or more data), making it possible to focus on larger scale training on a single component at a time. As we discuss in the next section, we believe the scalability to more complex environments can improve with larger-scale models - decoupling training does not affect the scalability of the models.
>
> **Scalability.** We agree that MiniGrid is a controlled setting. This is why we also evaluate on CSGO, which exhibits high-entropy first-person motion, long occlusions/re-appearances, and large viewpoint changes - failure modes typical of real video. Our goal is long-horizon content consistency, which goes beyond maximizing single-frame sharpness. The SSM state is a compact summary by design, while a diffusion decoder recovers visual detail each frame. We alotted most of our limited model compute budget for a lightweight diffusion model, complementing it with a cost-effective SSM. Under a larger compute budget, scaling up the SSM (number of parameters, state size, heads) and the diffusion model (training iterations, more parameters, newer model) is the most straightforward way to improve the visual quality and the capacity of the context preservation, including for more complex environments. For completeness, the final supplement will also show random rollouts (non forward-backward), clarifying that the low quality of the baseline results is most pronounced in the hardest, long-context cases.
>
> **Limitations.**
>  In Sup.Mat. Sect. 3.2 we have discussed our limitations and attributed them: (1) artifacts after strong action, attributed to the diffusion baseline; (2) decay of context details, attributed to the lossy SSM state (potentially scaling up the SSM can help). As CSGO is a pre-collected dataset of human play, it is difficult to separate clean subsets of failure modes in order to collect failure rate statistics. We can however comment that the decay seems to increase with longer-range dependencies. We can also demonstrate this decay on the predictions of MiniGrid context generalization up to context size 150, which is in contrast to the stable performance for context size up to 50 shown on Fig. 7(b). We show the average PSNR in ranges of 5 for the extended prediction:
>
> | PSNR (mean) |  1–5  | 6–10  | 11–15 | 16–20 | 21–25 | 26–30 | 31–35 | 36–40 | 41–45 | 46–50 | 51–55 | 56–60 | 61–65 | 66–70 | 71–75 |
> | :---------: | :---: | :---: | :---: | :---: | :---: | :---: | :---: | :---: | :---: | :---: | :---: | :---: | :---: | :---: | :---: |
> |             | 35.55 | 32.30 | 32.02 | 31.75 | 31.31 | 31.00 | 30.69 | 30.42 | 30.16 | 29.87 | 29.61 | 29.45 | 29.15 | 29.04 | 28.94 |
>
> **Contribution Importance.**
>  We agree that, at a high level, leveraging longer history should help. Our contribution is to make this practically effective for diffusion world models: a compact SSM state that carries long-horizon information with constant per-frame latency/memory, an architecture that feeds that information to a diffusion model, and a training scheme that allows for the diffusion model to make use of this information. Naively training end-to-end led to conditioning under-use. While others have used fixed image caches and explicit representations like pointclouds, this is the first work that explores exploiting a state space model for the benefit of a diffusion world model and makes it possible in practice. We consider this a significant contribution. We will improve our methodology and introduction sections to include these points more clearly.
>
> We also contribute a forward–backward evaluation with sufficient motion to test long-horizon consistency. Under this protocol, diffusion-only baselines often predict incorrect content, whereas our model reinstates previously seen content - showing the benefit is not just using more history, but how it is summarized and used under realistic compute. We further observe (see Sup.Mat. Sect. 3.2, Sect 3.6, context generalization in this rebuttal) robustness to large motion, generalization to much longer sequences, and that diffusion renders higher-quality frames later in the rollout than the SSM while still relying on SSM features.
>
> **Evaluation Clarifications.** We justify the split in Sup.Mat. L111-119. MiniGrid is an environment where an action results in a fixed amount of motion over 1 step. In MiniGrid, each action induces a fixed 1-step motion, so single-frame prediction cleanly isolates context recall from rendering quality. In CSGO, motion varies and unfolds over many frames; one-step prediction mostly probes short-context and both methods look strong. However, multi-step prediction highlights long-horizon consistency. Our model preserves context while the baseline hallucinates, so we adopt multi-step evaluation for CSGO and include MiniGrid multi-step qualitatives in Sup.Mat. Fig. 4.
>
> In CSGO multi-step rollouts accumulate camera/viewpoint drift: even with correct actions, slight motion mismatches de-align later frames (see Sup.Mat.). Content often matches while viewpoint/details differ. Thus PSNR/LPIPS fail to reflect content similarity, which we show experimentally in Sup.Mat. L127–137, Fig. 8. This is a known issue - see High-Resolution Image Synthesis with Latent Diffusion Models (p. 8); Image Super-Resolution via Iterative Refinement (p. 2). From Latent Diffusion: "simple image regression model achieves the highest PSNR and SSIM scores; however these metrics do not align well with human perception and favor blurriness over imperfectly aligned high frequency details." . Following common practice (EmuVideo/Stable Diffusion/Latent Diffusion), we therefore include human preference evaluations.
>
> The Simple MiniGrid setup is an easily reproducible minimal example designed entirely to showcase the the problem and our approach.  Even on a minimal color-recall task (two actions, almost no visual complexity, tiny dataset prone to overfitting), the diffusion baseline, limited by its context window, fails. Processing the long history with an SSM lets our model reliably recover the correct color. We then scale difficulty in MiniGrid and CSGO, showing that long-context recall persists as environment complexity rises.
>
> **Writing Suggestions.** We thank the reviewer for the detailed notes. We will add the requested clarifications on diffusion world model limitations (with the suggested citation), include a Data section figure illustrating MiniGrid and CSGO with example actions, and fix the noted typographical and caption issues.

---

> > ### Author Response · Authors · 2025-08-05
> >
> > We appreciate your review and would like to follow up to see whether our rebuttal has addressed the concerns you raised. If there are any remaining points that might benefit from further clarification, we’d be happy to provide additional input before the discussion period concludes. Thank you again for your time and consideration.

---

> ### Comment · Reviewer_1SP4 · 2025-08-05
> **a few additional questions**
>
> 1. Could you provide an answer to question 3, about line 247 in the paper: "We train and test on the same set of 34 samples, allowing for overfitting."? I'm not quite sure why this is needed/whether it's a limitation of the model.
> 2. Thanks for the clarification about end to end training. But I still don't see what the Diffusion model is doing, if the SSM is frozen during training. Isn't it simply learning a mapping from the SSM features to the image space? Is it not right to say that we expect the SSM features to implicitly represent everything pertaining to both scene representation and dynamics and the diffusion model is just operating on those features? Or to put it in another way, what exactly is the diffusion model doing that is not already done implicitly by the SSM?
> 3. How does this method relate to the diffusion forcing approach used in the recent paper on "Long-Context State-Space Video World Models", (Po, et.al.)?
> 4. Would the authors agree that scaling the SSM training to large scale data can be challenging? If not, could you provide more discussion on why you think this is straightforward? The Po, et.al paper I shared above seems to find that scaling SSMs is hard because of the fixed dimensionality of the latent state, and specialized architectural advances are needed to get them to work. So, I don't think it's right to say that the performance of your model is not limited by the SSM, given that joint training is not possible. I think that's a significant limitation, but I'm happy to have some more discussion on it. I think Reviewer VVHr has also raised a similar question.

---

> > ### Author Response · Authors · 2025-08-07
> >
> > ### 1.
> > To expand our response from Evaluation Clarifications, the small set **does not imply any model limitations**. Simple Minigrid is a minimal case without walls, a fixed navigation path, and a single fixed marker. We only vary the marker color - **one of 34 colors available in our Minigrid, resulting in 34 samples**. The color is then the only information to recall, making this a minimal reproducible test case. We explicitly acknowledge that the set is small (prone to overfitting) and this poses no problem for our results and conclusions of this experiment. DIAMOND, underusing history, cannot solve this minimal task, while StateSpaceDiffuser can. Larger experiments - MiniGrid with millions of layouts/markers and CSGO with millions of 3D environment frames, show the same advantage in settings where overfitting is impossible, so the minimal example does not reflect a model limitation.
> > ### 2.
> > In our framework the two branches are deliberately specialized.  **SSM is tasked with long-context compression**. At each time-step it receives a low-dimensional frame embedding, updates a compact recurrent state, and is trained with a next-embedding matching loss (MSE - non-variational). Its decoded RGB is therefore **blurry, with artifacts and less detail, but it encodes long-horizon scene information**.  **Diffusion is tasked with rendering.** It sees the pixels of the recent few frames and the current SSM state, and learns to **synthesize the pixels of the next high-quality frame**. During sampling it **fuses local visual cues** (from recent pixels) with the **global context** (from the SSM state).  Sup. Mat. Fig. 11 illustrates the division of labour: SSM-only outputs are with low visual quality yet with the correct context; Diffusion+SSM restores the high visual quality and preserves that context.
> > ### 3.
> >  (Po et. al.) is an independent from us work, made available as a preprint (26.05) after the NeurIPS submission deadline (15.05).
> >
> > Consistent with our premise, they also reinforce our hypothesis that integrating SSMs into a diffusion model yields superior long-range memory. They tackle this from a different angle and we find it complementary to our method. Their SSM operates on image tokens, embeds the SSM inside the diffusion backbone, and relies on diffusion-forcing with a clean-prefix trick, whereas we pass one whole-frame feature per step, keep the SSM as an external conditioning memory, and leave the diffusion objective unchanged.  Our approach does not allow conflating spatial/temporal patterns and leaves diffusion visual quality independent from the SSM - only the context is affected. (Po et al.) represent the same content with longer sequences and a possible diffusion bottleneck when SSM capacity is small but gain SSM integration with latent features.
> >
> > ### 4.
> > **Scaling an SSM is largely a matter of engineering details rather than an inherent SSM limitation**. SSM's **state size is a hyperparameter** that can be directly selected before training. Mamba has a hardcoded CUDA-related state size cap of 256. In Mamba-2, as used by (Po et. al.), there is no such cap and larger state sizes are allowed. We expect state size to increase scalability based on the original Mamba paper (Gu et. al.): "Of particular note is the dramatic improvement ... when the state size 𝑁 is increased". We are unaware of works showing the opposite.
> >
> > Instead of scaling up the state, (Po et. al.) took a different valid approach to address their specific architectural needs. They operate on image tokens and perform many SSM steps per frame (while we perform a single step of 256 tokens representing a full frame). However, computational complexity is $O(S^2)$, and increasing the state can compound for many steps (while still having the computational benefits over transformers - check Po et. al. Table 1). In addition, the long sequences entangling spatial/temporal tokens is more complex to learn patterns from and is linear in inference, slowing down the overall inference. Therefore, they split the tokens into patches (size $K$) and parallelize the process into separate SSMs with lower state size and shorter sequences, with computational complexity $O(K\cdot S^2)$.
> >
> > Our model, by contrast, applies one SSM step per frame (all 256 tokens), so compared to the token-per-step approach, increasing $S$ has only a marginal throughput impact. Thus we can scale state size directly (alongside number of parameters, layers, heads, etc.). In addition, as the methods are complementary, it is possible for the method of (Po et.al.) to be implemented in our Long-Context Branch.
> > Besides, our SSM is not built in the diffusion backbone, does not pose as a bottleneck for the diffusion architecture and does not affect visual quality - it only affects context quality. To confirm this, we again refer to Sup.Mat. Sec. 3.6 Fig. 11.
> >
> > Thanks for your response. Please let us know if we can further clarify any aspect.

---

> > > ### Comment · Reviewer_1SP4 · 2025-08-07
> > > **Thanks for the clarifications**
> > >
> > > I think you've addressed most of my concerns. Especially point number 2 was helpful. You might want to edit the discussion in the paper to make this more clear. The function of the Diffusion model was not very clear to me when I first read it.
> > >
> > > I understand that the (Po et al.) work was posted to arXiv after the NeurIPS deadline, and I don't expect a rigorous empirical comparison. However, as other reviewers have pointed out, the paper currently lacks a clear positioning within the broader context of related work. Even if a direct comparison isn't feasible, I believe a discussion that situates this work more explicitly among existing approaches would be valuable to the community.
> > >
> > > On point 4, I also agree with what you said. While it's primarily an architectural design choice aimed at scaling SSMs, it’s important to acknowledge that the current architecture may not trivially scale to more complex settings. As you mentioned, other papers have addressed this through various techniques like patch-wise SSMs. I think the paper would benefit from a transparent discussion of these limitation--both what challenges might arise in scaling the current setup, and what potential directions exist for overcoming them.
> > >
> > > I don’t think it's a problem that the results are on relatively simple environments--as long as the paper clearly communicates how it relates to prior work and honestly discusses current limitations and possible extensions. This might address the concerns of the other reviewers as well to some extent.
> > >
> > > Let me know your thoughts--I’m happy to continue the discussion a bit further before finalizing my rating.

---

> ### Author Response · Authors · 2025-08-08
>
> Thank you for the feedback. We will make sure to edit the Methodology section to express more clearly the purpose of the two branches.
>
> ## Positioning among other works in the field
> Upon the reviewer suggestion, **we plan to update our related work to more clearly position our work among others by including the works and suggestions from all reviews/discussions (incl. other reviewers) in this rebuttal**. In addition, we offer some extra insights on our positioning below:
>
> In our paper we have **tied our work to diffusion world models and state space models** by comparing with DIAMOND - a strong diffusion world model for smaller scale compute; and State Space World Model - a state space model adapted as a world model.
>
> In terms of grounding, we consider the closest work in literature to be S4WM - a model we discuss in main text in Line 39, Lines 126-128 (reference duplication will be fixed). It is a method based on the older S4 state space model that shows promise that SSMs preserve long context in video. Their model is entirely built on an SSM and tested entirely with an agent (no visual fidelity), on simple setups (remember the colors on a single line, or the colors of the walls in a 3D environment with a fixed layout). We were initially excited to compare our independent State Space World Model (our SSM-only model), with this method, and even try it out as our Long-Context Branch. However S4WM's code is not available and the authors did not respond to us. Therefore, we instead performed an ablation study with S4 in the main paper, showing that our State Space World Model, with the Mamba backbone performs better. While this is not a direct comparison with S4WM, the use of S4 bears similarity.
>
> On Lines 116-120 in the main paper we have shortly **discussed other recent world model long context methods** available at the time, and the relation to our work. To discuss a bit further, those methods are cache-based(storing input images, image representations, etc.) - to handle progressively increasing sequence of high-dimensional data, the cache has to explicitly select/delete sequence elements. Our method differs by forming a compressed representation of the history inside an SSM, capable of keeping information from each frame while throwing away the parts not useful for the current or future states. On Lines 129-134 **we position ourselves among other Transformer/Diffusion + SSM methods** (which we also plan to extend with discussion).
>
> ## Limitations and Scalability Discussion
> Our results confirm the method’s value under our compute budget. However, like the reviewer, we see the practical value of scaling up the model in future work for more complex environments, under a larger compute budget (Line 125 Sup.Mat.). We will include **a section in Sup.Mat. that makes explicit our current model's limitations under the scale we have and instructions on different directions to scale up the model**, as discussed in previous comments - by directly scaling model parameters (available in our specific architecture), and other specialized approaches, like parallelizing SSMs (Po et. al.). To be more explicit - **in the main paper text we will include comments in the experiment discussions with references to that section**. To further improve transparency about the model's limitations, we will include a **clear discussion in the main paper on limitations and strengths** with the main points already made in Sup.Mat as well as in this rebuttal, referring to the respective Sup.Mat. section for more details.
>
> We believe to have addressed all the questions - **relation to prior work**, and **limitations/extensions discussion**. Thank you for the constructive suggestions. We would appreciate your comment and assessment.

---

> ### Comment · Reviewer_1SP4 · 2025-08-09
> **That was really helpful**
>
> Thanks for the clarifications. That addresses all my concerns. One thing I would also suggest you is to move the Sup. Mat. Fig. 11 and the associated discussion to the main paper because I think that makes it easier to understand the method,  thanks!
>
> I will improve my rating.

---

### Official Review · Reviewer_eFWm · 2025-07-02

**Clarity:** 3
**Significance:** 2
**Originality:** 3
**Rating:** 3
**Confidence:** 4

**Summary:**

Existing world models often face insufficient memory issues due to the intractable increase in computation. This paper addresses this challenge by introducing Mamba, which aggregates all historical actions and frames with linear complexity. Conditioned on the global state extracted by Mamba, a two-stage diffusion model is employed to predict the next frame. The model shows superior long-term consistency in both the MiniGrid maze environments and the CSGO benchmark created by the authors.

**Questions:**

Q1) It is unclear how the model handles various context lengths at test time. Does the model always use a fixed context length during training? If the maximum training length is 50, can the model generalize to longer contexts in a zero-shot manner?

**Ethical Concerns:**

["NO or VERY MINOR ethics concerns only"]

**Final Justification:**

I thank authors for their efforts to address my concerns. The detailed rebuttal has partially resolved my questions. However, I still believe the evaluation are too simple and not generic enough, and the paper lacks a comparison with some published baselines to let the audience have a sense of the true advantages of this method. I am leaning rejection.

**Limitations:**

Please see the Weaknesses part.

**Paper Formatting Concerns:**

There are no major flaws in the formats.

**Quality:**

2

**Strengths And Weaknesses:**

# Strengths

S1) The paper aims to address a very challenging and intriguing topic in this field.

S2) Both quantitative evaluations and a careful user study are conducted for a comprehensive assessment of the proposed model.

S3) The writing is clear and easy to follow.

# Weaknesses

W1) **Computation concern still exists.** If my understanding is correct, the computation cost still grows linearly with the context length. How will the model manage context in real time when dealing with hundreds of frames?

W2) **Limited baselines.** The proposed model is compared only with a few baselines, primarily variants of itself. It would be more convincing to include some state-of-the-art methods, such as Diffusion Forcing (NeurIPS 2024), in the comparison.

W3) **Inferior qualitative results.** From Figure 6, the visualization after 17 frames looks quite blurry. Could you discuss potential reasons and solutions for that?

W4) **Duplicated references.** References 20, 21 are cited twice in the submitted manuscript. Please fix it in the revision.

---

> ### Author Rebuttal · Authors · 2025-07-31
>
> We thank the reviewer for their invaluable feedback and for acknowledging the importance of our topic, the comprehensiveness of our evaluation, and the clarity of our writing. We address the reviewer's comments below.
>
> **Computational Cost.** In inference, the state space model maintains a state in a streaming fashion, which allows for live processing of a growing sequence. At every time step of the sequence, the state is updated, used to generate the output, and kept for the next step. As we only need to keep the updated state, the latency remains constant - memory complexity and computational complexity per step remain the same and is not dependent on the sequence size. In contrast, transformers and CNNs (diffusion backbones) have increasing latency per step - even with KV-caching, the computational and memory complexity for a single step is $O(T)$, where T is the runtime. The complexity per step quickly becomes computationally infeasible with the growing context, making it necessary to use a cache of a few recent frames, preventing long-term dependencies. Conditioning the world model on the SSM state therefore preserves long-horizon information while keeping both memory and per-frame compute constant.
> In addition, we remind that while training, transformers still have a quadratic complexity, which is not the case with SSMs.
>
> **Baselines.** We thank the reviewer for suggesting Diffusion Forcing. It is a strong transformer diffusion model that excels at long video generation. However, we find that it too has the same context limitation, innate to transformers. We used their official online demo and performed a long context test on all 16 of the images they made available to try (Demo 3 - Image -> Extremely Long Video). In our setup we do a 60 degree turn in place to the right, followed by a 60 degree turn to the left. This resulted in 10 generated frames. In all of our 16 tries we observed the content from the start that was hidden by panning the camera to be forgotten - in its place completely new content is hallucinated at the end of the generated video. We are not allowed to upload images or add links, but we encourage the reader to attempt this quick test.
>
> Diffusion Forcing is targeted on long output generation, while we target consistency with long input (context) - even if generating a short future sequence. Our experiments with DIAMOND give us conclusions about the long-context abilities of our model. However, we recognize that, in future work, combining both paradigms can create a more capable model - both handling long generation and long context.
>
> **CSGO Qualitative Results.**  (1) As part of StateSpaceDiffuser, we train a light diffusion world model, under a fixed compute budget. As a result, frames in the long rollouts can occasionally appear with lower quality. Our contribution - the long-context mechanism via the SSM state, is decoder-agnostic: if a newer/larger conditional diffusion model is substituted, or if training time and sampling budget are increased, visual sharpness can improve without changing our method. Orthogonally, scaling the SSM (state dimension/heads and number of parameters) reduces the decay of high-frequency details carried over many steps, which also aids perceived sharpness. (2) Our primary objective is content consistency at long horizons. Therefore, the qualitative panels on Fig.6 and Sup. Mat. are in the challenging setting of forward-backward evaluation, with ensured enough motion to hide and reveal content. In these cases, the baseline produces inconsistent results while our method recalls the previously seen content, with a visual quality trade-off on the output. We will include qualitative examples in the final version of Sup. Mat. to demonstrate that the visual quality is better in the general case - on random sequences, without mirroring the action sequence (forward-backward).
>
> **Context Length.** In training, we always use training sequences of a fixed size. At a first glance, it may seem that this could lead to overfitting and the reviewer rightfully raises the question about generalizeability on different context lengths. StateSpaceDiffuser is capable of operating on much longer context lengths than it has been trained on, without any fine-tuning required. We have demonstrated this in Sup. Mat. Sect. 2.1 (Lines 50-56), where we show that a model, trained exclusively on context length 16, significantly outperforms the baseline on context length 50 - without any finetuning. To expand this discussion, as the reviewer asked, we take our model, trained on context length 50 and test it on much longer context sizes. To this end, we generate a new Minigrid test set with sequence length up to 150. Results are shown below:
>
>
> | Model                           | Avg. PSNR | Fin. PSNR | SSIM  |
> | ------------------------------- | --------- | --------- | ----- |
> | Context Length 100              |           |           |       |
> | Baseline (DIAMOND)              | 26.386    | 26.240    | 0.947 |
> | State Space World Model (Mamba) | 31.648    | 30.885    | 0.957 |
> | StateSpaceDiffuser (SSW)        | 37.993    | 35.866    | 0.977 |
> | Context Length 150  |   |     |   |
> | Baseline (DIAMOND)      | 24.346    | 24.197    | 0.938 |
> | State Space World Model (Mamba) | 27.927    | 26.980    | 0.935 |
> | StateSpaceDiffuser (SSW)        | 30.750    | 28.930    | 0.962 |
>
>
> It can be seen that StateSpaceDiffuser successfully generalizes to hundreds of frames in the context.  We observe visually reasonable visual and context recall quality - examples will be included in Sup. Mat.
>
> **Writing feedback.** We thank the reviewer for pointing out the double reference issue, which we will make sure to fix in the final manuscript.

---

### Official Review · Reviewer_dCL4 · 2025-07-02

**Clarity:** 2
**Significance:** 2
**Originality:** 2
**Rating:** 3
**Confidence:** 4

**Summary:**

To tackle the visual consistency problem for long sequences in diffusion world models, the paper proposes StateSpaceDiffuser with the aim to compress the interaction history into some latent space via a state-space model. Briefly, this consists of state-space model that operates over long sequences and learns to predict $\hat{f_t},… \hat{f_{T+1}}$ and the last four steps from this predicted are concatenated with action embeddings and passed as input to the diffusion model.  To evaluate temporal consistency by measuring how well can re-instantiate seen content on extended rollouts in 3 settings.

**Questions:**

To better understand this paper, it would be good to have a clear exposition of the following:

1.	What is the computational cost of the space-space model augmentation? It would be good to normalise this with the performance boosts in SSIM and PSNR. Particularly because confidence intervals are not reported due to computational constraints. However, it is celebrated repeatedly in the paper that the improvements are at essentially no additional computational costs.

2.	Given that the longest context window being used is 50 – I would like an evaluation with context window of 100 and 150 at the minimum. This would help understand at what point is the prediction appropriate?

3.	For evaluation, it would be good to understand if the model is trained for context of 50 – how much further can the model be unrolled (example 10 steps more) and then go back in sequence to the starting point.
4.	There seems to be research with human subjects but identified as N/A in the question. > ‘In CSGO we perform a user study, more aligned to the visual complexity of the environment’
5.	Somewhere in the paper, we need a clear definition of the agent for the minigrid settings. Currently, the policy is specified as visit 40 markers in a sequence. What does that entail exactly?
6.	The action embedding is separate in the generative (512) and long-context (16) branch? What is the reason for this?
7.	Have you tried adding noise in the middle of the sequence and evaluated how robust is the method for this?

**Ethical Concerns:**

["NO or VERY MINOR ethics concerns only"]

**Final Justification:**

I appreciate the author's efforts in introducing additional experiments and evaluations during the rebuttal phase. I have increased my score to reflect this (2-3).

**Limitations:**

Yes, discussed in detail in supplementary.

**Paper Formatting Concerns:**

There is a weird citation at the start of the reference, which doesn't exist - some others have the wrong year.

**Quality:**

2

**Strengths And Weaknesses:**

Strengths: The paper addresses a known limitation of diffusion-based world models i.e., their inability to maintain visual consistency over long sequences due to short context windows. To tackle this, it presents a rather straightforward approach to condition the diffusion model with state-space model features. The key strength of the paper is identifying an appropriate pipeline for training this; separate out the SSM training via Cosmos tokeniser features, and then load the pre-trained weights to train the diffusion. The evaluation protocol is well-designed, particularly the forward-backwards navigation setup that directly tests model capacity on MiniGrid and CSGO environments. The paper reports improvements on the baseline with 39.68 dB PSNR on MiniGrid for context length 50, and positive user study results on CSGO. Additionally, I appreciate the ablation studies to investigate how much the SSM is helping.

Weaknesses: The paper feels like an application contribution of combining two known techniques (MAMBA and diffusion) for learning WMs, with limited evaluation. For starters, the evaluation setting, although a useful starting point, is quite contrived and needs to be evaluated with 1. Long-time horizon, 2. one or two more environments with different levels of visual details, e.g, NetHack,  3., across more seeds to show reliability in the results presented. Furthermore, the CSGO evaluation relies on subjective user studies rather than objective metrics

---

> ### Author Rebuttal · Authors · 2025-07-31
>
> **Computational Complexity.** Mamba is known to have a linear complexity proportional to the sequence length, while trainsformers - a quadratic complexity over the (token) sequence length. While Mamba maintains a state, transformers and CNNs don't. To process the full sequence it has to be stored in its entirety in memory and processed at once. Even with KV-caching, the resulting linear complexity is only available in inference, and is linear per call - it does not support a streaming over a constantly increasing sequence. This is the case for current world models, including DIAMOND - a diffusion-based world model.
>
> Measurements of the computational cost of the SSM and the diffusion model in our particular case can be seen at Lines 25-29 in The Supplementary Materials. Our SSM takes only 0.6\% of all inference computations of the full model, for sequence size 16, and 1.8\% for sequence size 50. To account for this computational cost in our StateSpaceDiffuser scores, we normalize them by multiplying by $1-0.006$ for sequence size 16 and $1-0.018$ for sequence size 50. We report the normalized scores and compare with the baseline:
>
> | Model                     | Avg. PSNR (Norm.) | Fin. PSNR (Norm.) | SSIM (Norm.) |
> | :------------------------ | ----------------: | ----------------: | -----------: |
> | **Context Size 16**       |                   |                   |              |
> | DIAMOND                   |             27.13 |             25.44 |        0.950 |
> | StateSpaceDiffuser (Ours) |             34.41 |             37.81 |        0.974 |
> | **Context Size 50**       |                   |                   |              |
> | DIAMOND                   |             26.13 |             25.15 |        0.950 |
> | StateSpaceDiffuser (Ours) |             27.62 |             29.75 |        0.943 |
>
>
> Even after normalizing, significant gains are observed.
>
> **Generalization Over Longer Context.** We train on fixed-length sequences, which reasonably raises the reviewer's generalization concerns. StateSpaceDiffuser operates on much longer contexts without finetuning: in Sup.Mat. Sect. 2.1 (L50–56), a model trained only at length 16 already outperforms the baseline at 50. At the reviewer’s request, we further evaluate a model trained at 50 on lengths 100 and 150 using a new MiniGrid test set (up to 150). Results are below:
>
>
> | Model                           | Avg. PSNR | Fin. PSNR | SSIM  |
> | ------------------------------- | --------- | --------- | ----- |
> | Context Length 100              |           |           |       |
> | Baseline (DIAMOND)              | 26.386    | 26.240    | 0.947 |
> | State Space World Model (Mamba) | 31.648    | 30.885    | 0.957 |
> | StateSpaceDiffuser (SSW)        | 37.993    | 35.866    | 0.977 |
> | Context Length 150  |   |     |   |
> | Baseline (DIAMOND)      | 24.346    | 24.197    | 0.938 |
> | State Space World Model (Mamba) | 27.927    | 26.980    | 0.935 |
> | StateSpaceDiffuser (SSW)        | 30.750    | 28.930    | 0.962 |
>
> It can be seen that StateSpaceDiffuser successfully generalizes to 100 and 150 frames in the future, keeping a significant gain over the baselines. These results could be further improved by finetuning on longer sequences.
>
>
> **Evaluation Across Seeds.**
> As the reviewer suggested, we perform evaluation multiple times under different seeds. We run 4 times with different seeds for MiniGrid with context size 16 (computationally cheaper) and obtain $41.00\pm0.008$  Avg. PSNR, $40.52\pm0.019$ Fin. PSNR, $0.98\pm0.0004$ SSIM. We also perform evaluation over 4 seeds of a larger scale more expensive evaluation - our new 100 context length experiment from above. This results in: $37.99\pm0.002$ Avg. PSNR, $35.88\pm0.029$ Fin. PSNR, $0.98\pm0.000004$ SSIM. In both cases, the obtained metrics are extremely stable and consistent in between seeds (low std).
>
> **Evaluation Over Varying Visual Complexity.**
> In our main paper, we evaluate on 3 different environment setups with increasing level of complexity - the very constrained Simple MiniGrid (Line 245), free navigation in a maze (Line 257), 3D first-person environment (Line 274). We keep the model scale (hyperparameters) the same across these environments and evaluate the performance on each. To further compare the performance of StateSpaceDiffuser across environments with different visual complexities, we generate 2 more variants of our MiniGrid dataset based on visual complexity. We define complexity as number of markers and complexity of the maze walls (values in the range $[1,5]$). We generate a dataset low complexity (200 markers, difficulty 3), and one with high complexity (450 markers, difficulty 5). Our original MiniGrid dataset lies in between in terms of complexity (360 markers, difficulty 4). We take our StateSpaceDiffuser pretrained model on context length 50 (middle complexity) and evaluate it on the new datasets with varying difficulties (without any finetuning):
>
> | Model                            | Avg. PSNR | Fin. PSNR | SSIM |
> | :------------------------------- | --------: | --------: | ---: |
> | Baseline (low complexity)    |     26.09 |     25.60 | 0.95 |
> | Baseline (middle complexity) |     27.27 |     26.70 | 0.94 |
> | Baseline (high complexity)   |     23.09 |     22.87 | 0.93 |
> | Ours (low complexity)            |     36.72 |     35.78 | 0.97 |
> | Ours (middle complexity)         |     39.68 |     39.32 | 0.98 |
> | Ours (high complexity)           |     31.67 |     30.87 | 0.97 |
> It is observed that our model is better able to generalize to lower complexity rather than higher complexity. Still, in all cases, the performance remains higher than the baseline (DIAMOND).
>
> **User Study.**
> CSGO is characterized by actions with variable motion unfolding over multiple frames. Long rollouts accumulate small motion mismatches into camera-view drift, so later frames need not match ground-truth pixels. As shown in Sup.Mat. (L127–137, Fig. 8), content often remains similar while viewpoint/details differ, and PSNR/LPIPS under-report this similarity. This is a known issue - see High-Resolution Image Synthesis with Latent Diffusion Models (p. 8); Image Super-Resolution via Iterative Refinement (p. 2). From Latent Diffusion: "simple image regression model achieves the highest PSNR and SSIM scores; however these metrics do not align well with human perception and favor blurriness over imperfectly aligned high frequency details.". Following common practice (e.g. EmuVideo, Stable Diffusion, Latent Diffusion), we therefore include human preference evaluations.
> We thank the reviewer for pointing out the mistake in the question's response. The exact instructions, shown to our volunteers is available on Line 150-156 of Sup.Mat.
>
> **MiniGrid Exploration Policy.**
> The policy for generating the rollouts in MiniGrid is deterministic. We choose a sequence of 40 random markers (colors) and find the shortest valid path between them in the maze. This results in a very long path to follow - the actions are deterministic. Once 50 actions are complete into this path, the policy stops following the rest of the path and retraces the steps back to the beginning. This constitutes one rollout in our dataset. We will clarify this in our methodology section.
>
> **Separate Action Embeddings.**
> Our training consists of two separate stages: 1) training the Long-Context Branch; 2) training the Generative Branch, conditioned on the Long-Context Branch. Therefore, we keep separate action embeddings in each branch to be trained in each stage for the needs of the particular branch. In this way we are able to preserve the modularity - e.g. replacing the state space model without sabotaging any shared action embeddings. We keep the dimensions in the Generative Branch as chosen in the baseline (DIAMOND), for our Long-Context Branch we use more compact dimensionality (16) to account for the lower amount of parameters and computational budget for this branch.
>
> **Robustness to Noise.**
> We perform the experiment of adding noise in the middle section of the rollouts, suggested by the reviewer. In specific, we consider context length 50 in MiniGrid and add Gaussian noise (std 2.5) to the 11 frames in the middle (so noise remains symmetric in the sequence). We consider two cases: 1) adding noise to the SSM input only; 2) adding noise both to the SSM input and the diffusion model input. Results are shown below at different steps of prediction after the middle frame:
>
>
> | Model                 |    27 |    28 |    29 |    30 |    31 |    32 |    33 |    34 |    35 |    36 |    37 |    38 |    39 |    40 |    41 |    42 |    43 |    44 |    45 |    46 |    47 |    48 |    49 | 50    | 51    |
> | :-------------------- | ----: | ----: | ----: | ----: | ----: | ----: | ----: | ----: | ----: | ----: | ----: | ----: | ----: | ----: | ----: | ----: | ----: | ----: | ----: | ----: | ----: | ----: | ----: | ----- | ----- |
> | SSM noise   | 15.22 | 16.62 | 15.36 | 15.18 | 14.85 | 16.55 | 23.44 | 26.60 | 28.42 | 29.35 | 29.75 | 30.00 | 29.93 | 30.06 | 29.99 | 30.12 | 30.14 | 30.45 | 30.11 | 30.18 | 30.19 | 30.08 | 30.19 | 30.10 | 30.19 |
> | SSM + diffusion noise | 15.50 | 16.90 | 15.49 | 15.30 | 14.94 | 16.65 | 23.28 | 26.51 | 28.35 | 29.33 | 29.75 | 29.95 | 29.85 | 30.09 | 29.88 | 30.11 | 30.08 | 30.41 | 30.12 | 30.13 | 30.17 | 30.12 | 30.20 | 30.13 | 30.15 |
>
> We observe that for the specific frames with added noise, the performance decreases. We observe on those frames that content can disappear and long context is not correctly recalled. However, in both cases, within 4 steps after the noisy frames (after frame 34 - 5 noisy frames + window size 4) the memory and content recovers and is correctly predicted, with stable performance until the last frame. Still, the lower scores suggest some loss in performance. The fact that memory recovers after the noise suggests a certain level of robustness to noise.
>
> **Writing Suggestions.** Citation issues will be fixed in the final manuscript.

---

> > ### Author Response · Authors · 2025-08-05
> >
> > Thank you again for your review. We would like to follow up to ask whether our response has addressed your concerns. If there are any remaining points we could help clarify, we’d be happy to provide further input before the discussion period ends. Thank you for your time.

---

> > ### Comment · Reviewer_dCL4 · 2025-08-07
> >
> > Dear Authors,
> >
> > Thank you for your helpful answers and the new experiments - they are useful. Some questions I had:
> >
> > 1. Results with noise: these are very interesting because compared to Fig. 7B results, we have a decrease of 10 PNSR. Therefore, while there is robustness, there is a decline in the visual consistency. It would be good to have these results discussed in the limitations. Would humans be able to detect this degradation?
> > 2. For the evaluation on varying visual complexity, the baseline (DIAMOND) results are briefly mentioned but not included?
> > 3. The new comparisons on longer context are helpful - how much variability is in these results?

---

> ### Author Response · Authors · 2025-08-07
>
> We are glad our earlier responses were useful and appreciate the opportunity to clarify these remaining points.
>
> 1. The experiments and conclusions - both the memory robustness of our method and the decrease of the PSNR on the specific frames with added noise, will be reported in our final manuscript. The differences on the predicted frames with the noisy input are observable, not in a form of noise but in a form of missing visual elements (walls, markers). We believe the effect could be reduced by image augmentation strategies or engaging a more noise-robust pretrained tokenizer.
>
> 2. The baseline results are available in the first three rows of the table, provided for varying visual complexity in our previous response. The provided results and the extended analysis will be included in the final manuscript. Similarly to how we evaluate our method, we took the pretrained baseline DIAMOND on middle complexity, and we also evaluate it on the new low and high complexity datasets - the same that we evaluated our method (StateSpaceDiffuser) as well. To provide some extra analysis: StateSpaceDiffuser outperforms the baseline on each of the corresponding dataset for visual complexity. Between visual complexities, the baseline significantly drops in performance for higher complexity, and to a lesser amount to lower complexity - a similar pattern as with StateSpaceDiffuser. We will include those results in our final Sup.Mat. with a reference from the main text. If any additional analysis of our visual complexity experiment results is needed, please let us know.
>
> 3. **In terms of results stability/variability**, we have provided results for context size 100 in the "Evaluation across seeds" section of our response - the results are very stable as can be seen by the very low standard deviation across 4 different seeds. For completeness, we mention the results here: $37.99\pm0.002$ Avg. PSNR, $35.88\pm0.029$ Fin. PSNR, $0.98\pm0.000004$ SSIM. **In terms of dataset diversity/variability**, we generated our longer context test set (up to 150 context length) with middle complexity with the same setup and size as we have previously generated our original Minigrid dataset - with random wall layouts, marker positions and colors per rollout and the same navigation path policy.
>
> Thank you again for the constructive suggestions. We hope we addressed your remaining concerns, and we would be happy to answer any remaining questions.

---

> > ### Comment · Reviewer_dCL4 · 2025-08-08
> >
> > 1. Thank you - that helps clarify the decrease in PSNR. Is this effect consistently observed?
> > 2 + 3. Thank you for pointing these out.
> >
> > I am positive about the updated results and will update my score to reflect these changes.

---

> > > ### Author Response · Authors · 2025-08-09
> > >
> > > The results in the table are derived from evaluating on our entire test set,which makes the effect widely observable. In terms of visual results, we observed samples of a small subset and could observe the effect consistently.
> > >
> > > We have addressed all your questions. Thank you for the discussion.

---

### Official Review · Reviewer_VVHr · 2025-07-04

**Clarity:** 4
**Significance:** 4
**Originality:** 4
**Rating:** 5
**Confidence:** 4

**Summary:**

This paper tackles a key challenge in diffusion-based world models: the lack of long-term consistency in generated sequences due to their reliance on short observation windows. The authors propose StateSpaceDiffuser, a novel architecture that integrates a state-space model (specifically, Mamba) with a diffusion model to maintain a persistent memory of the environment throughout long rollouts. Experiments on both a synthetic 2D maze environment (MiniGrid) and a visually complex 3D environment (CSGO) demonstrate substantial gains in long-term fidelity over baseline diffusion models.

**Questions:**

Have you tried end-to-end training of the state and diffusion parts together, instead of training the state model separately first?

**Ethical Concerns:**

["NO or VERY MINOR ethics concerns only"]

**Final Justification:**

I maintain my recommendation to accept this paper. Long-term consistency is a crucial and underexplored challenge in world models. While the area is still in its early stages, this paper makes a meaningful contribution by proposing a novel architecture that demonstrates promising capabilities in long-term memorization, as evidenced by experiments on a Counter-Strike dataset. This is a valuable step forward and has the potential to inspire future work in this important direction.

After reading the other reviews, I feel that some of the critiques may be overly harsh given the scope and maturity of the problem. It is unrealistic to expect a single paper to solve such a fundamental and long-term research challenge in its entirety. While I acknowledge the current limitations—such as evaluation on a single domain-specific dataset—I believe the contributions are solid for a first step. These limitations can reasonably be addressed in future work.

**Limitations:**

They mentioned their limitations in supplementary material that the reconstruction quality is still not so good right now.

**Quality:**

4

**Strengths And Weaknesses:**

Long-term consistency is a very important problem for building world models, and the paper makes a solid first step toward addressing it. It studies the use of an external state model to maintain long-term memory, and the experimental results clearly show that this strategy helps. At first glance, it might seem like a straightforward idea, but I think the valuable part of this paper is that it successfully demonstrates the feasibility of this approach and opens up many interesting research directions—for example, how to improve the state features to contain more detailed information, or how to adapt the approach to more complex tasks beyond simply reversing actions.

One weakness is that the paper doesn't include a very extensive ablation on the choice of state model. I’m curious to see how simpler baselines like RNNs, LSTMs, or even vanilla transformers (e.g., LLMs) would perform. These comparisons could provide more insight into what properties of the state model really matter in this setup.

Missing citations:
Xiang et al. Pandora: Towards general world model with natural language actions and video states

---

> ### Author Rebuttal · Authors · 2025-07-31
>
> We thank the reviewer for recognizing the importance of our topic, experimental results and the potential for use in future models. We address the reviewer's comments below.
>
> **Long Sequence Models Ablation.**
> As the reviewer suggested, we have trained versions of the State Space World Model, but by replacing Mamba with LSTM and GRU for a context size 50 on our Minigrid setup. We add a linear layer on the input and output (dim 256, with ReLU activation) of the models. We are unable to use a standard transformer for a long image sequence because of the rising computational cost with each step - $O(N^2)$ complexity of the model in the general case. Furthermore, after trained, the transformer cannot generalize across a longer sequence than trained with. Results are shown below:
>
> | Model    | Avg. PSNR | Fin. PSNR |  SSIM |
> | :------- | --------: | --------: | ----: |
> | LSTM     |    26.80 |    26.906 | 0.93 |
> | GRU      |    27.40 |    26.680 | 0.93 |
> | Mamba    |    32.64  |   32.44  | 0.96 |
>
>
> Mamba outperforms the other sequence models, not having to perform BackPropagation Through Time (BPTT), among other advantages. In addition, as shown in Tab. 2 in the main paper, Mamba also outperforms S4 - another popular state space model. Compared to other methods, Mamba excels at modeling long-range temporal dependencies, while remaining not computationally demanding. This is confirmed in the original paper that proposed Mamba - "Mamba: Linear-Time Sequence Modeling with Selective State Spaces", where Mamba is compared with other state space models (Table 1) and transformer language models (Table 3).
>
>
> **End-to-end Training.**
>  We trained the SSM and the diffusion model end-to-end in two setups: (1) from scratch, and (2) with an SSM pretrained on sequence dynamics followed by joint training. However, the best results were achieved by first train the SSM to capture long-range structure and then freezing it while training the diffusion decoder.
>
> Joint training is challenging, as the components optimize on different timescales. The SSM needs many iterations of sequence-level supervision to stabilize a long-horizon representation, whereas the diffusion decoder optimizes a local denoising objective and rapidly finds a short-horizon solution that ignores a still-noisy SSM state (conditioning collapse). When we unfreeze a pretrained SSM for joint finetuning, decoder gradients quickly drift the SSM away from the representation that carries long-range cues; during this adaptation, the decoder again defaults to visual inputs and weakens reliance on the state.
> Freezing the SSM preserved its long-horizon information and led the diffusion decoder to make consistent use of it, yielding the strongest qualitative and quantitative results in our compute budget.
>
> We find that this approach of separately training the two components has the important benefits of drop-replacing one of them without finetuning the entire system. For example, it allows us to use the same diffusion decoder that we trained for context size 16 with a new state space model, that we trained for sequence size 50, and the new resulting StateSpaceDiffuser model works better for longer context, without requiring any diffusion training. (Note that even without training a new state space model, StateSpaceDiffuser can generalize to much longer context out of the box - check answer of Reviewer dCL4 for more details.)
>
> **Reconstruction Quality.**
> As part of StateSpaceDiffuser we train a lightweight conditional diffusion decoder under a fixed compute budget (no large pretrained backbone), which can yield frames artifacts late in long rollouts. Our contribution - the long-context conditioning mechanism via the SSM state, is agnostic to the choice of diffusion model. Replacing the decoder with a newer/larger one or training longer would improve visual sharpness without changing our method. Additionally, scaling the SSM (state dimension/heads/parameters) is expected to reduce high-frequency decay over many steps. Importantly, our primary objective is content consistency at long horizons. The qualitative results shown come from a forward–backward experimental setup with sufficient motion to occlude and later reveal content. On these challenging cases the baseline (DIAMOND) often becomes inconsistent or predicts incorrect content, while our model recalls the previously seen content - at some quality cost. For completeness, in the final Sup.Mat. we will also include qualitative examples from randomly sampled test sequences without the forward–backward constraint to show that visual quality in the general case does not degrade.
>
> **Writing Suggestions.**
> We thank the reviewer for the suggested work - we will add it to our related work.

---

> > ### Author Response · Authors · 2025-08-05
> >
> > Thank you again for your thoughtful and positive review. We’d be glad to hear whether our response reinforced your initial assessment or addressed any remaining questions. We really appreciate your engagement throughout the process.

---

> > ### Comment · Reviewer_VVHr · 2025-08-06
> >
> > Thanks for your reply! It answers many of my confusion. I will keep my original score -- that is, I lean toward acceptance.

---

### Note · Authors · 2025-08-12

We thank the reviewers and the AC. We have addressed all reviewers' concerns, with no remaining questions at the end of the discussion. The paper will be updated accordingly. Reviewers indicated positive rating/updates (dCL4,VVHr,1SP4) or updated rating without discussion/indication (eFWm).

We have addressed with detailed clarifications and discussion all questions about SSM computational complexity, qualitative fidelity, end-to-end training, SSM/Diffusion branch roles, user study, importance of our contributions. As reviewers asked, we have experimentally confirmed and analyzed: our longer context generalization; performance gains relative to GFLOPs; SSM performance relative to GRU/LSTM; reliability of our method under many seeds; robustness against varying visual complexity (low/medium/high); robustness against noise (drop and fast recovery); reviewer-suggested baseline Diffusion Forcing properties (lacking long-context). We have clarified all questions on methodology and evaluation details - Simple Minigrid as a minimum reproducible recall test; Minigrid data collection policy; reason for separate action embeddings; reason for CSGO multi-step evaluation. We have discussed model capabilities and limitations under the current computational budget, and concrete scaling paths, including for the SSM. We added detailed discussion to position our work relative to the reviewer-cited literature (transformer-based LCT; SSM-based Po et. al. (released after NeurIPS deadline)). We will implement all writing suggestions.

Paper changes include: Methodology and data clarifications, environments illustration, strengthening our positioning in the related work, discussions on computational complexity, limitations, advantages, scalability. Sup.Mat. Fig. 11 - to the main paper, adding the new experiments, adding Pandora citation, fixing duplicate references, suggested writing changes.

---

### Decision · Program_Chairs · 2025-09-17

**Decision:**

Accept (poster)

**Comment:**

This paper addresses the lack of long-term memory in diffusion-based world models. It introduces StateSpaceDiffuser, a hybrid model that uses a State-Space Model (SSM) to maintain a compressed history of an environment, which then conditions a diffusion model for high-fidelity frame generation. Experiments in 2D and 3D environments demonstrate that this approach significantly improves long-term visual consistency and recall compared to a diffusion-only baseline.

The paper's strengths are clear: it tackles a critical problem in generative modeling, proposes a novel and effective hybrid architecture, provides strong empirical evidence of its success, and uses a well-designed evaluation protocol to directly test its core claim of long-term recall. Initial weaknesses concerning evaluation depth and methodology were comprehensively addressed during the rebuttal with new experiments. The primary remaining limitation, noted by one reviewer, is the scope of the evaluation environments, suggesting that scalability to more complex, real-world domains is an important direction for future work.

For these reasons, I recommend acceptance. The paper makes a solid, novel contribution to a significant, unsolved problem. The authors rebuttal was critical to solidify the paper's quality, with new experiments that directly addressed initial reviewer concerns regarding evaluation depth and methodology. These should be incorporated in the camera ready. While the evaluation scope can be expanded, the current work provides a valuable proof of concept and opens a compelling new research direction. The AC also agrees with the positive reviewers, that given the maturity of this area, the progress demonstrated in this paper is sufficient for publication, despite the limited evaluation score.